# The Study on Molecular Profile Changes of Pathogens via Zinc Nanocomposites Immobilization Approach

**DOI:** 10.3390/ijms22105395

**Published:** 2021-05-20

**Authors:** Agnieszka Rogowska, Viorica Railean-Plugaru, Paweł Pomastowski, Justyna Walczak-Skierska, Anna Król-Górniak, Adrian Gołębiowski, Bogusław Buszewski

**Affiliations:** 1Centre for Modern Interdisciplinary Technologies, Nicolaus Copernicus University, Wileńska 4, 87-100 Torun, Poland; AGA4356@wp.pl (A.R.); rviorela@yahoo.com (V.R.-P.); pomastowski.pawel@gmail.com (P.P.); annkrol18@gmail.com (A.K.-G.); adrian.golebiowski@doktorant.umk.pl (A.G.); 2Department of Environmental Chemistry and Bioanalytics, Faculty of Chemistry, Nicolaus Copernicus University, Gagarina 7, 87-100 Torun, Poland; walczak-justyna@wp.pl

**Keywords:** zinc oxide nanoparticles, immobilization, antibiotics, kinetic study, MALDI-TOF MS, antibacterial activity

## Abstract

The most critical group of all includes multidrug resistant bacteria that pose a particular threat in hospitals, as they can cause severe and often deadly infections. Modern medicine still faces the difficult task of developing new agents for the effective control of bacterial-based diseases. The targeted administration of nanoparticles can enhance the efficiency of conventional pharmaceutical agents. However, the interpretation of interfaces’ interactions between nanoparticles and biological systems still remains a challenge for researchers. In fact, the current research presents a strategy for using ZnO NPs immobilization with ampicillin and tetracycline. Firstly, the study provides the mechanism of the ampicillin and tetracycline binding on the surface of ZnO NPs. Secondly, it examines the effect of non-immobilized ZnO NPs, immobilized with ampicillin (ZnONPs/AMP) and tetracycline (ZnONPs/TET), on the cells’ metabolism and morphology, based on the protein and lipid profiles. A sorption kinetics study showed that the antibiotics binding on the surface of ZnONPs depend on their structure. The efficiency of the process was definitely higher in the case of ampicillin. In addition, flow cytometry results showed that immobilized nanoparticles present a different mechanism of action. Moreover, according to the MALDI approach, the antibacterial activity mechanism of the investigated ZnO complexes is mainly based on the destruction of cell membrane integrity by lipids and proteins, which is necessary for proper cell function. Additionally, it was noticed that some of the identified changes indicate the activation of defense mechanisms by cells, leading to a decrease in the permeability of a cell’s external barriers or the synthesis of repair proteins.

## 1. Introduction

The control of pathogenic microorganisms growth is extremely important for public health and food safety [1,2]. According to the World Health Organization (WHO) [3], the greatest threat is associated with drug-response bacteria such as *Pseudomonas*, *Acinetobacter* and *Enterobacteriaceae* (e.g., *Escherichia coli*, *Klebsiella*). The recent data (report under the European Centre for Disease Prevention and Control from 2021) [4] indicate that, despite the recognition of growing antibiotic resistance as one of the priorities of health policy in the European Union, this problem is still one of the most important challenges for public health. The dissemination of the described phenomenon is related to several reasons, among which too frequent or unjustified use of antibiotics can be distinguished. According to the WHO, European countries are in the forefront of the highest intake of antibiotics. Antibiotics such as ampicillin and tetracycline are usually the first-line choice in the treatment of many infections [5,6]. Therefore, despite the unreasonable use of the listed drugs and the known ampicillin/tetracycline-resistance of pathogenic strains, it is necessary to study these strains and antibiotics in the context of searching for a new type of antiseptic. Due to the huge increase in bacterial infections caused by drug resistance strains, modern medicine and researchers have a promising task for developing new alternative agents for the effective control of bacterial diseases. In this regard, zinc and its various forms have attracted great interest from researchers. One strategy of zinc is its use as an essential biogenic element for the proper functioning of living organisms [7,8,9]. Another strategy is its non-toxicity for eukaryotic cells, unlike silver. However, many studies show the presence of toxic effects in the blood by the penetration of zinc ions to the prokaryotic cells [8,9]. Additionally, many researchers report that antibacterial effect of zinc salts (e.g., zinc chloride, zinc acetate) against various bacteria and fungi strains depends on both contact time and Zn^2+^ concentration [8]. Zinc chloride activity was found against *E. coli* [10,11], or zinc acetate against *Staphylococcus aureus* and *Pseudomonas aeruginosa* [12]. It is also necessary to underline that the absence of microorganism resistance to this element was recorded [7]. Moreover, the presence of properties such as a lack of taste, smell and low cost allows for the use of this element as an ingredient in cosmetics, medicines or as a preservative in food products. [9]. Furthermore, it was also demonstrated that zinc oxide nanoparticles seem to have even more interesting properties and greater antibacterial effectiveness. They are multifunctional inorganic nanoparticles with unique chemical, optical, semiconductor and pizoelectric properties, and their antibacterial activity against both pathogenic and spoilage organisms was proven in numerous studies [1,2,7,13,14,15]. Such individuals are characterized by an increased specific surface area, as the small size of the particles determines the increase in their reactivity. In addition, zinc oxide nanoparticles have an effect on photocatalysis and photooxidation in biological and chemical species and, at the same time, it is bio-safe material [2]. Due to the many possible methods of zinc oxide nanoparticles synthesis (e.g., sol-gel, co-precipitation, hydrothermal, biological) and the control of the synthesis process, it is possible to obtain nanoparticles with various properties (morphology, shape, size) and different antibacterial potentials [1,9]. Interestingly, our previous works show that the combination of nanoparticles with conventional pharmaceutical agents (e.g., antibiotics) might be a great opportunity for commonly used treatments [16,17,18]. It leads to a synergistic action—a phenomenon occurring as a result of combining two different factors in order to increase their individual activity [19]. Moreover, the immobilization of ampicillin/tetracycline, considered ineffective against strains such as *S. epidermidis* or *K. pneumoniae* on nano-ZnO, is a novel and promising approach to enhance drug activity. The synergistic effect of the antibiotics-ZnONPs complex was confirmed in numerous studies [20,21,22,23]. A study by Gupta et al. [24] indicates the synergistic effect of streptomycin and biologically synthetized zinc oxide nanoparticles using *Catharanthus roseus* leaf extract against six pathogenic bacteria, including *P. aeruginosa* and *E. coli*. In addition, Banoee et al. [25] showed that the application of zinc oxide nanoparticles in combination with ciprofloxacin resulted in an increase in the inhibition zones of 27% and 22% against *S. aureus* and *E. coli*, respectively. However, they also reported that such a combination with amoxicillin, penicillin G and nitrofurantoin resulted in a decrease in antimicrobial properties against *S. aureus*.

Moreover, the mechanisms of the antibacterial action of zinc ions, zinc oxide nanoparticles and their combinations with antibiotics are still under discussion, and continue to be controversial [2]. In the case of zinc ions, it was assumed that actions such as inhibiting active transport as well as disrupting the enzyme system along with affecting the metabolism of amino acids, is responsible for their antibacterial effects [2]. In turn, the mechanism of the antibacterial action of zinc oxide nanoparticles seems to be more complex [8]. Many researchers indicate that it consists of three main factors: (i) the induction of oxidative stress, (ii) the release of free zinc ions and (iii) direct contact with cell surface [1,8,9]. The photocatalytic generation of reactive oxygen species (ROS) exhibiting oxidizing properties and high reactivity leads to the destruction of numerous bacterial cell building components (e.g., DNA, proteins, lipids) [2,9]. In turn, direct contact of nanoparticles with the cell surface may result in the loss of integrity of its wall and membrane [2]. It is believed that the antibacterial effect of zinc oxide nanoparticles can be associated with adsorption on the cell surface or the accumulation inside the cell, which ultimately results in the disruption of cellular activity and a change in the permeability of cell membranes [2]. What is more, these nanoparticles, having an abrasive surface texture, can mechanically damage the bacterial wall and membrane. This is evidenced by studies carried out by Ramani et al. [26,27], which showed that the antibacterial activity of zinc oxide nanoparticles depends on their surface, which in turn depends on their shape. Similarly, the mechanism of the synergistic action of metal nanoparticles and their oxides in combination with antibiotics is still unknown. To date, this phenomenon was attributed to the role of nanoparticles as drug carriers, or to a different mode of action of both factors [28]. Therefore, there is a strong need for further research to understand the mechanisms underlying the antibacterial properties of zinc based substances. Accordingly, in this study, an interdisciplinary approach, including a commonly used microbial assay (MIC), along with complementary experimental techniques (flow cytometry, spectroscopy and mass spectrometry) is proposed. It will allow us to connect physiological changes not only at the cellular level but also at the molecular level.

Therefore, the aim of this study is to describe the molecular profile changes of selected pathogens via a zinc nanocomposites immobilization approach. The work was divided into a few stages: (i) an immobilization of the commercially available chemically synthetized zinc oxide nanoparticles with two antibiotics (tetracycline, ampicillin), and an investigation of the antibiotics’ mechanism bindings (effectiveness and nature of sorption) onto ZnO nanoparticles; (ii) a determination of the antibacterial potential of immobilized and non-immobilized zinc oxide nanoparticles, tetracycline, ampicillin and zinc ions against *Escherichia coli*, *Staphylococcus epidermidis* and *Klebsiella pneumoniae*, using a minimum inhibitory concentration (MIC) and flow cytometry study; (iii) investigations of the changes in molecular bacterial profiles (proteins and lipids profiles) after the antibacterial agent treatments using the Matrix-assisted laser desorption/ionization time-of-flight mass spectrometry technique (MALDI-TOF MS).

## 2. Results and Discussion

The non- and immobilized ZnONPs were first subjected to an investigation of the mechanism of zinc oxide nanoparticles’ immobilization with antibiotics. Secondly, the zeta potential and FTIR were studied for a physico-chemical characterization using DLS. Next, an investigation of antimicrobial properties, lipids and protein profile changes followed. All studies were performed for the ZnONPs (non-immobilized), ZnONPs/TET and ZnONPs/AMP (immobilized) nanoparticles. The Zn^2+^ ions, TET (tetracycline) and AMP (ampicillin) were used in all the experiments for the sake of comparison.

### 2.1. Investigation of the Mechanism of Zinc Oxide Nanoparticles’ Immobilization with Antibiotics

In order to determine the nature of the ampicillin and tetracycline sorption process on the surface of zinc oxide nanoparticles, kinetic studies were performed. Figure 1A,B shows the kinetic of the sorption process as a graph of antibiotics concentration versus time for ampicillin and tetracycline, respectively, while Figure 1C,D presents the effectiveness of the ampicillin and tetracycline sorption process onto zinc oxide nanoparticles. The obtained curves reveal that the antibiotics’ sorption on the surface of chemically synthetized zinc oxide nanoparticles exhibit a complex and complicated nature. Moreover, the mechanism of the sorption strictly depends on the structure of the antibiotics. In the case of the ampicillin sorption, the process consists of three different stages. The first one is related to the rapid sorption process, the second one results from the gradual sorption, and the final step is the sorption equilibrium. The first stage of sorption ends after 60 min of incubation and, at that time, 80.21 ± 2.57% of the antibiotic is sorbed. The second stage begins at 60 min and ends with 480 min of incubation, followed by the establishment of an equilibrium. The maximum capacity of the ampicillin sorption onto zinc oxide nanoparticles amounts 810.63 ± 3.18 mg/g, and it allows for a bound of over 99%. For tetracycline, the sorption mechanism is different. In this case, the process consists of only one very fast sorption stage, ending after 10 min of incubation, followed by an equilibrium. The effectiveness of this antibiotic binding process is also different. The maximum sorption capacity of zinc oxide nanoparticles was only 671.59 ± 0.87 mg/g, which allowed the binding of 70.73 ± 0.07% of the antibiotic.

For the linear fragments of the process, the rate constants were calculated based on the zero order kinetics model, which is appropriate for the description of every sorption step characterized by the linear relationship. The obtained values of the rate constant are summarized in Table 1, and they constitute a real physical parameter determining the speed of the observed sorption stages. In the case of ampicillin, the rate constants amount to 2.5372 and 0.0853 (mg/L)/min in the first and second stages, respectively. In turn, for tetracycline, the calculated value of the rate constant was 14.3326 (mg/L)/min.

The pseudo-first and pseudo-second order kinetic models were also applied to present obtained kinetics data. Figure 1E,F shows plots of fitting these models to experimental data, and Table 1 summarized the appropriate rate of kinetics constants. The determined values of relative approximation errors (A_approx._) show that, in the case of both antibiotics, the pseudo-second order kinetic model is more suitable for the description of the sorption process. Moreover, it can be assumed that the pseudo-first order kinetic model is more accurate for the description of the ampicillin sorption than tetracycline, and, conversely, the pseudo-second order kinetic model ensures a better fit for tetracycline.

Furthermore, in order to describe more precisely the nature of antibiotics sorption by zinc oxide nanoparticles, the Weber–Morris intra-particle diffusion model was applied. Figure 1G,H presents the dependence of zinc oxide sorption capacity on ampicillin and tetracycline in the function of the square root of time. The results show that, for ampicillin, the sorption process consists not only in the surface sorption but also in the sorption in the intradiffusion layer. On the other hand, in the case of tetracycline, the antibiotic binding occurs only on the surface of nanoparticles. To determine the thickness of the external sorption surface (A), as well as the value of the intra-particle diffusion coefficient (K_ip_), the *y*-axis intercept and slope of the second sorption step line were calculated. The volume of the external sorption surface amounts to 574.577 and 628.753 mg g^−1^, while the intra-particle diffusion coefficient was 10.959 and 0.263 mg g^−1^min^−0.5^ for ampicillin and tetracycline, respectively.

The obtained data of the zero order kinetics rate constants indicated that the sorption process occurs more abruptly for tetracycline. However, despite faster sorption of tetracycline at the beginning of the process, its maximum efficiency was definitely lower than in the case of ampicillin. This is due to different mechanisms of antibiotic binding by zinc oxide nanoparticles. In the case of tetracycline, sorption occurs only on the surface of nanoparticles and, after its saturation, equilibrium occurs. It is different in the case of ampicillin where, after the initial stage of the surface sorption and the saturation of the nanoparticles surface, the antibiotic diffuses into the interior of the nanoparticle. As a result, the total sorption capacity of ampicillin is higher, and it is allowed to bind almost all of the antibiotics present in the solution.

In addition, the Gibbs free energy (ΔG^0^) and distribution coefficient (K_d_) of the process of antibiotics binding on zinc oxide nanoparticles (ZnONPs + AMP = ZnONPs/AMP or ZnONPs + TET = ZnONPs/TET) were calculated (Table 2). For ampicillin, these values amount to −31.542 kJ/mol and 3.8 × 10^5^, respectively. In turn, for tetracycline, it was −22.293 kJ/mol and 8.9 × 10^3^. The negative values of Gibbs free energy obtained for the sorption process in the case of both antibiotics indicate their spontaneous nature. Furthermore, the lower value obtained for ampicillin proves greater process spontaneity than in the case of tetracycline.

### 2.2. Physico-Chemical Characteristics of Immobilized Zinc Oxide Nanoparticles

To determine the influence of zinc oxide nanoparticles’ immobilization with selected antibiotics on their dispersion stability and hydrodynamic size distribution, a zeta potential measurements and dynamic light scattering (DLS) analysis was performed. Table 3 summarizes the values of zeta potential and the average hydrodynamic size of each nanoparticle population in a function of the pH in which the immobilization process was conducted (physiological conditions, pH ~7). It was noted that the antibiotics adsorption by nanoparticles, in the case of both ampicillin and tetracycline, did not significantly affect the electric charge of tested individuals. Moreover, the zeta potential value indicates the stability of dispersion for all of the investigated samples (ζ < −20 mV) [16]. However, significant changes were observed in the size of the distribution profile of nanoparticles after the immobilization process. In the case of non-immobilized nanoparticles, three size populations can be seen. Therefore, the hydrodynamic radius distribution of unmodified nanoparticles exhibit polydisperse in nature. According to the manufacturer’s information, their actual size measured by transmission electron microscope (TEM) is below 100 nm. The observation of large sizes of nanoparticles using the DLS technique is the result of the fact that this method is based on the measurement of their hydrodynamic size. Therefore, the obtained results indicate the polydisperse nature of the hydrodynamic radius distribution of nanoparticles before and after functionalization, also explaining the aggregation ability of the particles.

Figure 2 show the quantitative share of each population. Nanoparticles with a size of almost 900 nm had the largest share. Particles of the size of about 200 nm had a smaller significance. The third population, which was observed as being over 5000 nm, may be associated with the aggregation of nanoparticles. In turn, after nanoparticles immobilized with ampicillin, only two signals were observed (~700 nm; ~200 nm), and for those immobilized with tetracycline, only one was observed (~400 nm). In the case of tetracycline-functionalized nanoparticles, the distribution of the hydrodynamic radius shows a Gaussian character. However, the observed signal is too broad to consider the sample as a monodispersed—it is related to low resolution of DLS technique when measurements are performed using only one scattering angle. Therefore, it indicates the polydisperse nature of the sample.

Another approximation of particle size is the radius of gyration (R_g_). This approximation is the root mean square distance from the axis of rotation of different particles. These values are obtained using the multi-angle light scattering (MALS) technique. Determining the radius of gyration, as well as the molar mass distribution of biocolloids and biopolymers, is an important issue in the field of physicochemical description of nanoparticles [29,30,31]. The applied analytical method allowed us to obtain ZnO particle size values before and after immobilization with ampicillin and tetracycline. Figure 3 shows the fractograms from the separation of particles using the direct injection method (without focusing step) to asymetric flow field flow fractioantion (AF4) channel. With this, it was not possible to focus the fractions of larger particles (agglomerates). These particles only left the canal when purging the channel. In the fractogram from the analysis of ZnONPs, the presence of two populations of particles differing in the diffusion coefficient can be distinguished. The second fraction is the tail of the first fraction. The values of the R_g_ are given in Table 3. On this basis, the elution model of these particles can be established as normal (Brownian), where the separation occurs according to the increase in the particle diffusion coefficient. Both individual fractions of particles are highly monodisperse (Pdi very close to 1). After the ZnONPs were immobilized with antibiotics, only one particle fraction was recorded. The data on the mean radius of gyration show that the particles of ZnONPs with tetracycline are larger. The fractions in both cases are equally monodisperse. Considering the absence of further particle fractions after immobilization with antibiotics, it can be concluded that the obtained product was more monodisperse in terms of particle size. Amde et al. [32] performed the fractionation of ZnO nanoparticles using the AF4-MALS technique. They showed a great influence of the experimental conditions on the obtained measurement results. The authors also noted lower particle size values when performing the experiment with TEM techniques in relation to light scattering techniques (MALS and DLS), which are similar to the results obtained in our work.

The obtained results performed by MALS and DLS are complimentary. Both of the respective measurements provide information regarding the size particles populations and can generate more detailed size characterization. The DLS generated the information regarding the hydrodynamic radius of the sample based on the intensity of light scattered by a colloidal dispersion, while MALS measured the radius of gyration at light scattering at several different angles, based on molar mass.

Moreover, in order to confirm the antibiotics’ adsorption on the ZnONPs surface, the possible changes on the recorded FT-IR spectra were examined. The negative value of adsorption on the recorded spectra is related to the data processing algorithm of the used instrument. According to the technical guide regarding the analysis method for direct detection, the software subtracts buffer signal from only a part of the spectrum, resulting in a partially processed spectrum covering the region where subtraction has been applied. In the next step, the partially processed spectrum is integrated. The software anchors a baseline, that runs parallel to the x axis, at a basepoint outside the exact analysis region, and determines the strength of the amide signal at the predetermined wavenumber. Figure 4 presents the obtained FT-IR spectra for zinc oxide nanoparticles non-immobilized and immobilized with ampicillin and tetracycline. As the presence of characteristic bands on carbon was observed on the spectrum of unmodified, chemically synthesized and commercially available zinc oxide nanoparticles, an additional analysis was carried out to determine the total carbon (TC) content in the tested sample. The obtained results indicate that the TC of the sample is approximately 60 mg/L (organic carbon—49 mg/L, and inorganic carbon—11 mg/L). This element may be a residue from the synthesis process or come from a stabilizer [33,34]. However, after the nanoparticles’ immobilization with antibiotics, a few changes at the recorded spectra were noted. These changes include signal sharpening at 1380 cm^−1^ after nanoparticles’ immobilization with tetracycline. This signal can be assigned to δCO group vibrations. In addition, a signal shift of 1415 → 1425 cm^−1^ was observed after the zinc oxide immobilization with this antibiotic. It can be related to the bending vibrations of the CH_3_ group. Moreover, signal broadening in the range of 1550–1670 cm^−1^ was noted after the nanoparticles’ immobilization with both antibiotics, which can be attributed to υCC vibrations in the phenyl ring [35,36,37]. All the observed changes indicate ampicillin and tetracycline adsorption on the surface of the zinc oxide nanoparticles. Moreover, the changes in phenyl ring vibration may indicate the π-π interaction between zinc oxide nanoparticles and antibiotics molecules (Figure 5) [38,39]. It can be assumed that the interactions in the ZnONPS-drug complex exhibit a mixed nature. Hydrophobic interactions probably have a decisive contribution in the binding due to the hydrophobic nature of zinc oxide [40]. However, in accordance to the previous results concerning the interaction of zinc with organic ligands, water can also participate in these interactions, which is related to the solvation process [41].

### 2.3. Investigation of Antimicrobial Properties of Immobilized Zinc Oxide Nanoparticles

To determine the antibacterial potential of the immobilized zinc oxide nanoparticles, the minimum inhibitory concentration (MIC) was determined for non-immobilized and immobilized nanoparticles, along with antibiotics and zinc ions, against three selected bacteria: *E. coli*, *S. epidermidis* and *K. pneumoniae*. Table 4 summarizes the obtained MIC values for each tested variant. The results indicate that the zinc ions were the least effective against all the investigated bacteria strains compared to other tested antimicrobial agents. In turn, zinc oxide nanoparticles successfully inhibited the microorganisms’ growth.

Zinc oxide nanoparticles present the highest effectiveness in the case of *S. epidermidis* and the lowest for *E. coli*. Tetracycline also exhibits effective antibacterial action against all the strains, whereas ampicillin is not effective enough in the cases of *S. epidermidis* and *K. pneumoniae*. Moreover, the increase in the antibacterial activity of zinc oxide nanoparticles after their immobilization with TET in relation to *E. coli* was observed, while the immobilization with ampicillin did not affect the antiseptic effect for the respective bacteria strain. In turn, an increased antimicrobial effect against *S. epidermidis* was observed for ZnONPs/AMP compared to AMP alone, whereas ZnONPs/TET generated the same MIC value as in the case of ZnONPs before immobilization. As for *K. pneumoniae*, the immobilization of ZnONPs with both antibiotics did not affect their MIC value. In the case of *S. epidermidis* it should also be noted that the MIC value is quite high for AMP, whereas clinical strains of this bacteria are generally susceptible to this antibiotic at concentrations from 32 to 0.125 µg/mL [42]. It may be related to the environment from which the used strain was isolated. Honey is an extreme environment for bacteria due to the high content of sugars and natural antibacterial compounds, the presence of hydrogen peroxide as well as low pH and high osmotic pressure [43]. Therefore, growth under such conditions can generate the acquisition of new traits by cells, leading to increased resistance to commercial drugs [44,45]. This proves that the use of functionalized zinc oxide nanoparticles can effectively inhibit the growth of even more resistant strains.

In order to attain basic information about the mechanism of the action of tested agents on bacterial cells, a flow cytometry analysis (FC) was performed. The investigated concentrations of each antibacterial agent were consistent with the obtained MIC value. Figure 6 presents the percentage of live and dead cells of the selected bacterial strains and Figure 7 shows the fluorescence spectra of cells after treatment with the tested agents. The results show that, in the case of *E. coli* and *S. epidermidis*, all tested antimicrobial agents affect these cells in a similar way, both in the ratio of live/dead cells as well as in the action mode (Figure 6A,B and Figure 7A,B). In the case of Zn ions, the number of dead cells is definitely lower in comparison to ZnONPs, whereas, in the case of both tested antibiotics, a significant number of death cells were generated. Additionally, the florescence of the signals coming from disrupted bacteria cells fragments were also observed, and their number was higher in the case of tetracycline (Figure 7). The immobilization of zinc oxide nanoparticles with AMP leads to the generation of a smaller number of dead cells compared to the AMP alone (Figure 6). However, the ZnONPs/AMP exhibits a different mode of action in comparison to non-immobilized ZnONPs. This is evidenced by the observed changes in the fluorescence spectra of cells after flow cytometry analysis. In the case of unmodified nanoparticles, signals from live and dead cells were observed, while the functionalization of ZnONPs with ampicillin resulted in the appearance of broken cell fragments (Figure 7). The exception was *K. pneumoniae* cells, where only dead cells were observed after the application of both unmodified and ampicillin-modified nanoparticles. In turn, after the nanoparticles’ immobilization with TET, the presence of dead cells significantly increased compared to non-immobilized counterparts; a slight increase in the number of cell fragments was also observed. Compared to *E. coli* and *S. epidermidis*, *K. pneumoniae* presented a different trend of the antimicrobial agents’ action (Figure 6). The antibiotics’ action mechanism, in the case of *K. pneumoniae*, was similar to the ones of *E. coli* and *S. epidermidis* (Figure 7). However, for the other tested agents, in comparison to the results obtained for the rest of the bacteria, significant differences were observed (Figure 6 and Figure 7). In this case, zinc ions and ZnONPs were definitely more dead in *K. pneumoniae* cells, presenting an increased anti-bactericidal effect, destroying the cells by almost 90% in the case of Zn^2+^ and 100% in the case of ZnONPs (Figure 6). Moreover, the ZnONPs/AMP led to the death of almost all bacteria cells (Figure 6), but the fragments of destroyed cells were not observed on the spectra (Figure 7). Thus, it can be concluded that the mechanism action of ZnONPs against *K. pneumoniae* does not change after immobilization with AMP, whereas, in the case of ZnONPs/TET, a significant change in the mechanism of action could be noticed (Figure 7). In this case, the treated cells generated the fluorescence of live cells.

Moreover, based on MIC results, the increasing effect of the immobilized ZnONPs against *E. coli* seems to be generated by the counterparts alone. However, analyzing the generated fluorescence in the flow cytometry results, it can be seen that the mechanism of action of immobilized ZnONPs is dissimilar to the tested counterparts. In the case of ZnONPs/AMP and ZnONPs/TET, more differences were observed.

Many previous studies suggest that one of the dominant mechanisms of the antibacterial activity of zinc oxide nanoparticles is related to the zinc ions’ release into the solution [1,2,8,9]. However, our results obtained for *E. coli* and *S. epidermidis* show that, in the case of the respective bacterial cells, the release of free zinc ions does not significantly contribute to the antibacterial activity of nanoparticles. Only in the case of *K. pneumoniae* was the potential of zinc ions to kill bacteria cells observed, but the MIC value was still very high. Therefore, it can be assumed that, in this case, the antibacterial effect of zinc oxide nanoparticles is mainly associated with its direct contact with the cell surface. Similar conclusions were made by Brayner et al. [46]. Their results indicate that zinc oxide nanoparticles lead to the disorganization of *E. coli* membrane, which results in an increase in membrane permeability and the accumulation of ZnONPs in this membrane, along with the cellular internalization of NPs. Moreover, studies conducted by Ramani et al. [26,27] indicate that the antibacterial properties of zinc oxide nanoparticles depend on their structure and shape. Padmavathy and Vijayaraghavan [10] suggest that the antimicrobial properties of zinc oxide nanopartices can be related to the mechanical damage of the cell membrane due to their abrasive surfaces. This mechanism of antibacterial action of zinc oxide nanoparticles was also postulated by Stoimenov et al. [47].

There can also be found a number of studies that indicate the synergistic effects of zinc oxide nanoparticles and antibiotics. Rubab et al. [7] show that the addition of zinc oxide nanoparticles to erythromycin, trimethoprim-sulfamethaxazole, ciprofloxacin and ceftriaxone increase the antibacterial activity of these antibiotics against the range of pathogenic bacteria strains. Additionally, the results presented by Sharma et al. [19] confirmed the synergistic effect of ZnONPs’ combination with ciprofloxacin against *E. coli* and *S. aureus*. Namasivayam et al. [23] studied the effect of the combination of ZnONPs with ofloxacin, norfloxacin and cephalexin against *S. aureus*, *E coli* and *P. areuginosa*. All tested formulations pointed to the synergy of both factors. In our study, based on the flow cytometry results, the synergistic effect was noticed only for ZnONPs/TET in relation to *E. coli* (Figure 7). Fattah et al. [48] show that the addition of zinc oxide nanoparticles does not always increase the effectiveness of antibiotics. They observed only a slight increase in the inhibition zone after the use of zinc oxide nanoparticles in combination with amoxicillin and cefotaxime against an antibiotic resistant *K. pneumoniae* strain, as well as after the application of aztreonam and amoxicillin against an antibiotic resistant strain of *E. coli*. In turn, no synergistic effect was observed for both strains after using nanoparticles in combination with ampicillin and tetracycline. Similarly, in our study, the MIC values for those non-immobilized and immobilized with these antibiotics did not change against *K. pneumoniae*. The MIC value was also the same for non-immobilized and immobilized ZnONPs with TET against *E. coli*. Moreover, in turn, according to the MIC value, in the case of ZnONPs/AMP against *E. coli*, a slight decrease in nanoparticle activity was noted. Based on the dissimilarity of generated fluorescence in the case of the flow cytometry analysis, it cannot be excluded that this phenomenon is connected with the different modes of action of the ZnONPs and AMP formulations when they are combined together. An agonistic effect of zinc oxide nanoparticles and some antibiotics was also noted by Abo-Shama et al. [49]. Nevertheless, a synergistic effect was observed for ZnONPs’ combination with cefotaxime, cefuroxime, fosfomycin, azithromycin, oxacillin and oxytetracycline against *E. coli*, along with cefotaxime, cefuroxime, azithromycin, fosfomycin, chloramphenicol and oxytetracycline against *S. aureus*. Its combination with most antibiotics results in the antagonistic effect against *Salmonella* spp.

### 2.4. Growth Kinetics of the Selected Bacteria Strains in the Presence of Tested Antibacterial Agents

A growth kinetics study was performed in order to determine the optimal time for microorganisms cultured in the medium supplemented with tested antibacterial agents. It was checked how it affects the growth of microorganisms. For this step, based on the MIC value, the optimal concentration (25% of the MIC value) was chosen. The antibacterial doses were selected experimentally and based on our previous studies with silver (bio)nanoparticles [18] to obtain the sufficient amount of the bacterial cells for the analysis by the MALDI technique. Figure 8 depicts the growth kinetics curves of *E. coli*, *S. epidermidis* and *K. pneumoniae* treated with AMP, TET, Zn^2+^, ZnONPs, ZnONPs/AMP and ZnONPs/TET, and non-treated cells that served as a control. In the case of *E. coli*, the application of zinc-based substances in such a concentration did not affect the bacterial growth rate, and a sharp increase in the cell number was observed after only nine-hour incubation. However, the cell growth in the presence of both antibiotics was definitely slower, even at a much lower concentration than the minimum inhibitory concentration. For *S. epidermidis*, cell growth after the application of all formulations was definitely underestimated more than it was for controls. The least impact on bacterial growth occurred in the case of ampicillin. After the AMP treatment, an increase in the cell number was observed after only 12 h of incubation. For other antibacterial agents, a slight increase in the cell number after this time was noted. The smallest increasing effect of all antiseptic factors was observed in the case of *K. pneumoniae* where, after 6 h of incubation, a rapid increase in the bacterial cell number was observed.

On the basis of the obtained results, it was found that the twenty-four-hour incubation of cells under such stress conditions was the optimal time to obtain a sufficient number of cells to perform a MALDI analysis for all tested variants.

### 2.5. Electrophoretic Analysis of Isolated Proteins from the Bacterial Cells after Treatment with Selected Antimicrobial Agents

To check whether the addition of tested antibacterial agents at concentrations of 25% of the MIC value affects the expression of bacterial proteins, gel electrophoresis of proteins isolated from treated *E. coli* cells was performed as a screening analysis. One-dimensional gel electrophoretic (1D-GE) analysis provides the first information on the changes taking place in the native protein profile while, in the case of the MALDI analysis, the post source decay of high protein complex can occur. Therefore, in this study, 1D-GE consists in a complementary method for the MALDI technique. The analysis was carried out as an example only for *E. coli* cells not treated and treated with ZnONPs, ZnONPs/TET, Zn^2+^ and TET. It can be noted that, for native bacteria cells, only two intense bands with masses in the range about 10–15 kDa appeared (Figure 9). In turn, after cells treatment with zinc-based substances, the appearance of many more bands was observed. Similarly, incubation with tetracycline led to the largest changes in the observed electropherogram. This result indicates the intensive effect of the tested agents at selected concentrations on the protein profile of microorganisms.

### 2.6. MALDI-TOF MS Analysis of Proteins and Lipids Extracted from the Bacterial Cells Treated with Selected Antimicrobial Agents

To obtain a more accurate view of changes occurring in cells cultured under selected conditions, a MALDI-TOF MS analysis was performed for proteins isolated from each of the tested bacteria. All noticed proteins profile changes in the recorded spectra were summarized and individual signals were assigned to probable corresponding bacterial proteins, according to the UniProt database. Cultured cells in the medium, with the addition of all tested antibacterial agents, led to numerous changes in the recorded spectra compared to the spectra obtained for cells not exposed to stress. The most changes in the protein profile were recorded in *E. coli* cells. One of the significant protein modifications for these bacteria include a signal loss at 2341 m/z after the incubation of the cells with both antibiotics. This signal can be attributed to the RNA helicase, which plays an important role in RNA processing and DNA replication, repair, recombination and transcription [50]. In addition, after using the immobilized ZnONPs, signals appeared at 3281 and 3324 m/z, indicating the presence of SOS protein mutagenesis and repair protein, along with the high-affinity branched-chain amino acid ABC transporter permease, responsible for repairing DNA damage and intermembrane amino-acids transport, respectively [51,52]. Similarly, another protein (5658 m/z—putative oligopeptide transporter) responsible for intermembrane transport also appears after the use of antibiotics. The presence of ZnONPS/AMP also causes the expansion of the protein responsible for zinc ions binding (6361 m/z—putative zinc-binding dehydrogenase) [53]. After the application of zinc ions, there are also signals from cold shock proteins (7320 m/z), acid stress (9065, 9714 m/z) and potassium ion transport (7352 m/z). In addition, after the incubation of the cells with TET, the presence of protein, causing cell membrane cracking, can be observed, which confirms the results obtained by flow cytometry (7964 m/z—lysis S family protein) [54,55]. In the case of *S. epidermidis*, a few changes in the molecular protein profile were observed, and most of the observed signals come from new, uncharacterized proteins. The changes in *K. pneumoniae* protein profile include, for example, the signal losses at 7354, 8933 and 8953 m/z after applying all tested factors. Their disappearance is probably associated with a lack of expression of putative urocanase, phosphate transport ATP-binding protein PstB and amidohydrolase, which are proteins important for the proper functioning of the cell and responsible for hydrolase activity. In addition, after the application of all test substances, the appearance of the transcriptional regulator RutR protein (5444 m/z) was observed, causing the transcription retention of transcription from the DNA [56]. Moreover, the signals that appear after the application of zinc-based substances come from the crossover junction endodeoxyribonuclease (7282 m/z), iron transporter (7768 m/z), cytoplasmic copper homeostasis protein cutC (7796 m/z) and cation transport regulator (8938 m/z). All these proteins are responsible for the binding and transport of metal ions. In addition, the crossover junction endodeoxyribonuclease is also responsible for the process of repairing damaged DNA.

In order to carry out a more accurate investigation of the changes occurring in the metabolism of bacteria treated with tested substances, an analysis of changes in their lipid profiles was performed. According to AlMasoud et al. [57], the lipids were extracted from cells using a mixture of chloroform/methanol (2:1). The MALDI-TOF MS spectra for lipids extracts were recorded in both positive and negative ion mode at the range of 190–2500 m/z. To examine whether the signals observed in the spectra were derived from lipids, a LIFT-TOF/TOF analysis was additionally performed for the most intense signals. Figure 10 presents an example of MS/MS spectra recorded in a positive ion mode of a protonated molecular ion, at 722.311 and 774.756 m/z, obtained from *E. coli* native cells. Performing the phospholipid fragmentation leads to the disruption of the phosphate-glycerol bond, thanks to which it becomes possible to identify lipids based on a signal from the characteristic head group of phosphatidyl. Therefore, at the MS/MS spectrum of 722.311 m/z (Figure 10A), the signal 163.948 m/z corresponds to the sodiated head group of phosphatidyl ethanolamine (PE) [(C_2_H_5_N)H_3_PO_4_ + Na]^+^. In addition, it can be concluded that molecular ion at 722.311 m/z derives from [PE (13:0/20:3) + Na]^+^, which also confirms the presence of the signals at 563.495 and 585.497 m/z, which come from the detachment of the sodiated head group and the head group of PE from this phospholipid, respectively. The remaining fragmentation ions in the spectrum also indicate the presence of PE. The ion at 120.921 m/z indicates the presence of [H_3_PO_4_ + Na]^+^ and the ion at 683.621 m/z comes from the detachment of the C_2_H_5_N group from the identified compound. Similarly, the molecular ion at 774.754 m/z can be identified as phosphatidyl glycerol (PG) [PG (14:0/22:2) + H]^+^. This is indicated by the presence of a signal from the sodiated head group of PG at 195.111 m/z [(HOCH_2_CHOHCH_2_)H_3_PO_4_ + Na]^+^ and a signal at 603.759 m/z, associated with the disconnection of PG head group from this lipid [58].

Furthermore, to determine the changes in the total lipid profile of bacteria after their exposure to ampicillin, tetracycline and zinc ions, along with zinc oxide nanoparticles non-immobilized and immobilized with antibiotics, spectra obtained for each variant were compared with those recorded for native bacteria. The observed changes in the signals with the assigned probable lipids were summarized and analyzed according to LIPID MAPS^®^ Lipidomics Gateway database. As can be seen, the addition of all the tested antibacterial agents to the medium resulted in numerous changes in the composition of bacterial lipids, which were manifested in the disappearance or appearance of signals on the spectrum compared to cells not exposed to stress. As in the case of protein profiles, the most changes were observed in *E. coli*. In addition, the most intense signal loss was observed as a result of the use of TET for all tested strains. Comparably few signals were also obtained for *S. epidermidis* treated with ZnONPs/AMP. As observed on the growth curves, this factor had the most impact on *S. epidermidis* growth inhibition, even at a concentration of 25% MIC value. This result may indicate the degradation of lipids building bacterial cell membranes and the loss of cell integrity as a result of exposure to these factors, as it was also described by the flow cytometry analysis, where the signals from cell fragments were observed. In addition, as a result of stress, an increase in the amount of free fatty acids in the cells of all tested strains was observed. It may also indicate the cell membrane degradation through the breakdown of long chain phospholipids. Moreover, in the case of *K. pneumoniae*, although many signals on the spectra of cells treated with antibacterial substances did not disappear, their intensity was definitely lower, which may also indicate the partial degradation of membrane lipids. On the other hand, the increased amounts of glycerophosphoglycerols (PG) and cardiolipins (CL) were also observed due to stress in all cells. According to Kuyukina et al. [59], the increased production of these lipids can lead to a decrease in cell envelope permeability. Thus, it can be assumed that this is one of the defense mechanisms activated by bacterial cells as a result of exposure to stress. Figure 11 summarizes the observed changes in bacteria protein and lipid profiles after their incubation with the tested factors.

Furthermore, a statistical analysis was performed to describe the statistical differences between all tested strains treated by a specific antibacterial agent. Figure 12 presents a dendrogram differentiating the treated cells depending on differences in their protein (Figure 12A) and lipid (Figure 12B) profiles. In the case of bacterial protein profiles, specific grouping of samples treated with factors from the same groups can be observed (Figure 12A). The spectra of proteins isolated from bacteria treated with antibiotics constitute one cluster, which is most closely related to native cells. The next distinct group are cells incubated with ZnO nanoparticles and Zn^2+^. The last cluster consists of cells treated with immobilized nanoparticles—cells treated in this way show the most differences compared to native bacteria. In the case of statistical differences between bacterial lipids profile (Figure 12B), it can be seen that the native cells are definitely different from the treated cells. It can also be observed that tetracycline apparently influences the lipid profile of the tested bacteria. Moreover, bacteria treated witch ZnO nanoparticles and tetracycline immobilized ZnO nanoparticles affect similar changes in bacterial lipid profiles. On the other hand, the nanoparticles after the immobilization with ampicillin increased the number of changes in the lipid profile, both in relation to the use of ampicillin and nanoparticles alone.

In this study, attention was paid to the impact of these agents on microbial metabolism. In the first stage, the immobilization of chemically synthesized zinc oxide nanoparticles with ampicillin and tetracycline was performed. The mechanism of antibiotics binding on the surface of nanoparticles was also determined by kinetics studies. The results showed the complex nature of this process specific to each antibiotic. The effect of the immobilization on the physicochemical properties of nanoparticles, such as size and surface charge, was also investigated. Then, the change in the antibacterial activity of nanoparticles after immobilization in relation to selected pathogens was assessed using flow cytometry and the determination of minimal inhibitory concentrations. An increase in nanoparticle activity was observed after immobilization with tetracycline against *E. coli*, and a decrease in its biological activity was seen after immobilization with ampicillin against *S. epidermidis*. Finally, the investigation in the bacteria molecular profile after its treatment with selected zinc-based antibacterial agents by the MALDI technique was performed. Numerous changes were shown in the protein and lipid profiles of microorganisms. These changes led to a loss of cell integrity due to membrane lipid degradation. Additionally, some identified changes indicating the activation of defense mechanisms by cells led to a decrease in the permeability of the cell’s external barriers or the synthesis of repair proteins.

## 3. Materials and Methods

### 3.1. Chemicals and Reagents

Antibiotics, zinc nitrate, zinc oxide nanoparticles and other chemicals (i.e., solvents for HPLC and MALDI-TOF MS, α-cyano-4-hydroxycinnamic acid, bacteria growth media) were obtained from Sigma-Aldrich (Merck, St. Louis, MO, USA). Ultra-pure water was obtained by Milli-Q RG system (Millipore Intertech, Bedford, MA, USA).

### 3.2. Kinetic Study of the Antibiotics Sorption onto Zinc Oxide Nanoparticles

The antibiotics (500 µg/mL) and zinc oxide nanoparticle (464 µg/mL) solutions, preliminary suspended in water, were mixed to a ratio of 1:1. Then, the samples were incubated at room temperature for a specific period of time with continuous mixing. Next, samples were centrifuged (RT, 10 min, 15,000 rpm) and the supernatant was analyzed by the HPLC technique (Shimadzu Prominence, Tokyo, Japan) to determine antibiotic concentration in the supernatant, according to a previously described methodology [16,17]. All samples were prepared and analyzed in triplicate.

The content of sorbed antibiotics onto zinc oxide nanoparticles, the effectiveness of the sorption process along with binding rate constants for each kinetic models were calculated, according to Buszewski et al. [16]. The amount of antibiotics bound by ZnO NPs (1), sorption effectiveness at each incubation period (2), binding rate constants for zero (3), pseudo-first (4), pseudo-second (5), order kinetic models and Weber–Morris intra-particle diffusion model (6) were calculated using the following equations:q_t_ = (C_0_ − C) V/m(1)
E% = 100 × (C_0_ − C)/C_0_(2)
C = C_0_ − k_0_ × t(3)
q_t_ = q_e_ (1 − exp(−k_1_ × t))(4)
q_t_ = q_e_^2^ × k_2_ × t(1 + q_e_ × k_2_ × t)(5)
q_t_ = A + K_ip_ × t^0.5^(6)
where: m—mass of sorbent (g); V—volume (L); C_0_—initial antibiotics concentration (mg/L); C—antibiotics concentration at different time period (mg/L); q_t_—amount of sorbed antibiotics at different time period (mg/g); E—sorption effectiveness %; t—time (min); k_0_, k_1_ and k_2_—rate constant of zero ((mg/L)/min), pseudo-first (min^−1^) and pseudo-second (g × mg^−1^min^−1^) order kinetics model, respectively; q_e_—amount of antibiotics sorbed at equilibrium (mg/g); A—constant that indicating the thickness of the boundary layer diffusion or external surface adsorption (mg/g); K_ip_—diffusion rate constant ((mg/g)/t^0.5^).

The distribution coefficient (K_d_) (7) and the change of Gibbs free energy (ΔG^0^) (8) for the antibiotics’ sorption by ZnO NPs was calculated from the kinetic data at an equilibrium by applying the equation:K_d_ = q_e_/C_e_(7)
ΔG^0^ = −RTlnK_d_(8)
where: R is the gas constant (8.314 J/mol∙K) and T is the adsorption temperature (295 K).

### 3.3. Immobilization Procedure

The equal volumes of tetracycline (TET) or ampicillin (AMP) solution (500 µg/mL) and zin oxide nanoparticles’ suspension (464 µg/mL) were thoroughly mixed, and then incubated at room temperature and shaken for 72 h. Then, the samples were dialyzed by the MWCO 3500 Spectra/Por dialysis membrane at room temperature for another 72 h. The functionalized/immobilized antibiotics with zinc oxide nanoparticle concentrations were measured by a CX 7500 Spectrometer ICP-MS.

### 3.4. Physico-Chemical Characteristics of Immobilized Zinc Oxide Nanoparticles

A total 2 μL of ZnONPs before and after immobilization was spotted onto a card and dried. Fourier transform infrared spectroscopy (FT-IR) analysis was performed in the range of 1350–1850 cm^−1^ using a direct detect spectrophotometer (Merck, Darmstadt, Germany).

The Zetasizer NanoSeries was used for the hydrodynamic size and zeta potential measurements. The samples were mixed, sonicated directly before analysis and then transferred to disposable folded capillary cells (Malvern Instruments, Malvern, Great Britain).

The content of total carbon in the samples of commercially available zinc oxide nanoparticles was measured using Total Organic Carbon Analyzer TOC-L (Shimadzu, Duisburg, Germany). Directly prior to the analysis, the sample of ZnONPs was diluted in ultra-pure water to obtain the zinc oxide concentration of 1 μg/mL.

ZnO nanoparticles before and after immobilization with Ampicillin and Tetracycline were subjected to particle size distribution analysis using the multi-angle light scattering technique (MALS). The AF2000 Multi Flow system (Postnova Analytics GmbH, Landsberg am Lech, Germany) was used in this research. A MALS detector PN3621 (Postnova, Germany) was used for data acquisition and set at 35 °C with 80% laser (λ = 532 nm) power. The signal was registered from 20° to 156° angles. The UV detector, as a concentration detector, acquired data at 364 nm. Water was used as a carrier liquid and prepared from the Milli-Q system (Merck Millipore, MA, USA). Before analysis, the carrier liquid was filtered using a 0.1 μm nylon membrane (Merck Millipore, Warsaw, Poland). The direct injection mode (without focusing step) was used in this study. The injection volume was 50 μL of 1 mg/L of sample. The detector flow was 0.4 mL/min. All fractionation analyses were performed at room temperature. Evaluation of the MALS signal was performed using AF2000 Control software. The experimental data were fitted through the sphere model. The polydispersity index was calculated as the ratio of the weight to number average R_g_.

### 3.5. Minimum Inhibitory Concentration (MIC) Assay

For this purpose, 10 concentrations (300, 150, 75, 50, 25, 12.5, 6.25, 3,125, 1.56 and 0.78 µg/mL) of zinc ions (Zn^2+^) and zinc oxide nanoparticles (ZnONPs) non-immobilized and immobilized with ampicillin (ZnONPs/AMP) and tetracycline (ZnONPs/TET) were tested against *Escherichia coli* ATCC25922, *Klebsiella pneumoniae* ATCC BAA-1144, purchased from Pol-Aura (Dywity, Poland), and *Staphylococcus epidermidis* isolated from honey [60] (accession number MH045861) from the collection of Centre for Modern Interdisciplinary Technologies, Nicolaus Copernicus. The MIC assay was performed based on the microdilution method using 96-well cell culture plates (PP) and Mueller-Hinton (MH) broth/medium, according to the Clinical and Laboratory Standards Institute (CLSI) guidelines, with a slight modification. The cultured bacterial cells (1 × 10^6^ CFU/mL) were mixed with prepared concentrations in a ratio of 1:1. Then, 12 μL (final concentration of 45.8 μL/mL) of in vitro toxicology assay kit, resazurin-based (Sigma-Aldrich, St. Louis, MO, USA), was added to each well in order to determine bacterial viability. Afterwards, the preparation samples, with resazurin as an indicator, were applied. At the same time, a negative control was performed for the nanoparticles in MH broth medium, without the addition of bacteria culture, to exclude the influence of nanoparticles on the color change of the resazurin. MIC value was determined according to Elshikh et al. [61], based on the changed resazurin color from blue to pink. However, the time of the samples’ incubation with the resazurin addition was established at 24 h in the dark at 37 °C, according to Sarker et al. [62]. Such a long time of incubation also enabled better modeling of the clinical action of the tested antibacterial agents, as even small amounts of remaining bacterial cells could lead to noticed reduction of resazurin. All measurements were prepared in triplicate. Bacterial cells without antimicrobial agents were used as a control.

### 3.6. Flow Cytometry Analysis

For the flow cytometry analysis, the samples were prepared in the same way as described in Section 3.5., with a suitable modification. In this case, resazurin was not used but a dedicated kit for the flow cytometry analysis [63] was employed. All samples were prepared according to the manufacturer’s protocol. The final concentration of each antibacterial agent in the well was based on the determined MIC value.

### 3.7. Growth Kinetics of the Selected Bacteria Strains in the Presence of Tested Antibacterial Agents

This step was performed in order to find the optimal incubation time for the MALDI analysis. The equal volume of cultured bacterial cells (1 × 10^6^ CFU/mL) and investigated formulations were mixed, then incubated for 24 h at 37 °C. The untreated cells served as a control. All of sample variants were prepared in triplicate. In order to examine the bacterial growth, absorbance was measured every three hours (starting from 0 h and ending after 24 h), using a Multiskan FC Microplate Photometer (TermoScientific, Vantaa, Finland) at λ = 600 nm.

### 3.8. Electrophoretic Analysis of Isolated Proteins from the Bacterial Cells after Treatment with Selected Antimicrobial Agents

The goal of the electrophoretic study was to screen whether the incubation of cells with the tested antibacterial agents had a significant impact on their protein profile. The electrophoretic analysis was performed using Bolt™ 4–12% Bis-Tris Plus Gels and 20X Bolt™ MES SDS Running Buffer (Novex Life Technologies, Carlsbad, CA, USA). For this step, the proteins extracted from non-treated and treated *E. coli* cells were investigated with selected antimicrobial agents (zinc ions, zinc oxide nanoparticles, tetracycline and zinc oxide nanoparticles immobilized with tetracycline), using the FOCUS™ Bacterial Proteome kit (G-Biosciences, St. Louis, MO, USA), according to the instructions provided by the manufacturer.

### 3.9. Samples Preparation for MALDI-TOF MS Analysis

The bacteria cells were prepared in accordance with Section 3.7, but in a larger volume. The obtained cultures were transferred to a new sterile falcon tube, centrifuged (10 min, 10 °C, 4000 rpm) and obtained pellets were washed twice with sterile water. Then, the protein and lipid extractions were performed according to Buszewski et al. [18].

### 3.10. Statistical Analysis

The statistical significance of differences in the lipid and protein profiles of *E. coli*, *S. epidermidis* and *K. pneumoniae*, after their incubation with zinc ions, ampicillin, tetracycline and zinc oxide nanoparticles, and before and after immobilization with antibiotics, was determined by Statistica MIC testus t.sta. Based on the presence or absence of characteristics signals recorded on the MALDI spectra, the clustering analysis was carried out.

## 4. Conclusions

This study presents, for the first time, the complexity of the process of antibiotic (AMP and TET) sorption on the surface of chemically synthesized zinc oxide nanoparticles. The sorption kinetics study shows that the antibiotics’ sorption on the surface of ZnONPs is a complex process and specific to each antibiotic structure. In the case of ampicillin, the sorption process consists of three stages while, in the case of tetracycline, there were only two stages. Moreover, the efficiency of the process was definitely higher in the case of ampicillin, where the process occurred very rapidly. The maximum capacity of ZnONPs for ampicillin and tetracycline achieved about 811 and 672 mg/g, respectively. In addition, the Weber–Morris intraparticle diffusion model reveals that tetracycline sorption is only a result of surface binding while, for ampicillin, the diffusion of antibiotic particles inside the structure of the nanoparticles was also noted. It was shown that immobilization leads to increased homogeneity of the hydrodynamic size of the nanoparticles, but it does not adversely affect their stability under physiological conditions. Although the ampicillin’s structure allows for a greater sorption on the surface of nanoparticles, its presence did not affect their antibacterial efficacy and, in the case of *S. epidermidis*, slightly reduced it. In turn, immobilization with tetracycline led to an increase in the activity of ZnO nanoparticles against *E. coli*. In addition, flow cytometry showed that nanoparticles after immobilization show a different mechanism of action. The application of immobilized nanoparticles resulted, in most cases, in an increase in cell fragments compared to unmodified nanoparticles (non-functionalized). Additionally, the MALDI-TOF MS proved to be an effective tool for tracking changes in lipid and protein profiles of pathogens treated with the investigated antibacterial agents. Non-immobilized ZnONPs and immobilized ZnONPS/TET affect changes in bacterial lipid profiles in a similar way. In turn, the ZnONPS/AMP increased the number of changes in the lipid profile, both in relation to the use of ampicillin and in the nanoparticles alone.

Changes in the protein and lipid profile spectra indicated that the mechanism of antibacterial activity of tested agents is mainly based on the destruction of cell membrane integrity by destroying lipids. In addition, there was a lack of expression of many proteins necessary for proper cell function. At the same time, some of the observed changes indicated the activation of cell defense mechanisms. These changes consisted of reducing cell membrane permeability by increasing the amount of glycerophosphoglycerols and cardiolipins. Moreover, the expression of proteins responsible for repairing damaged DNA was also noticed.

## Figures and Tables

**Figure 1 ijms-22-05395-f001:**
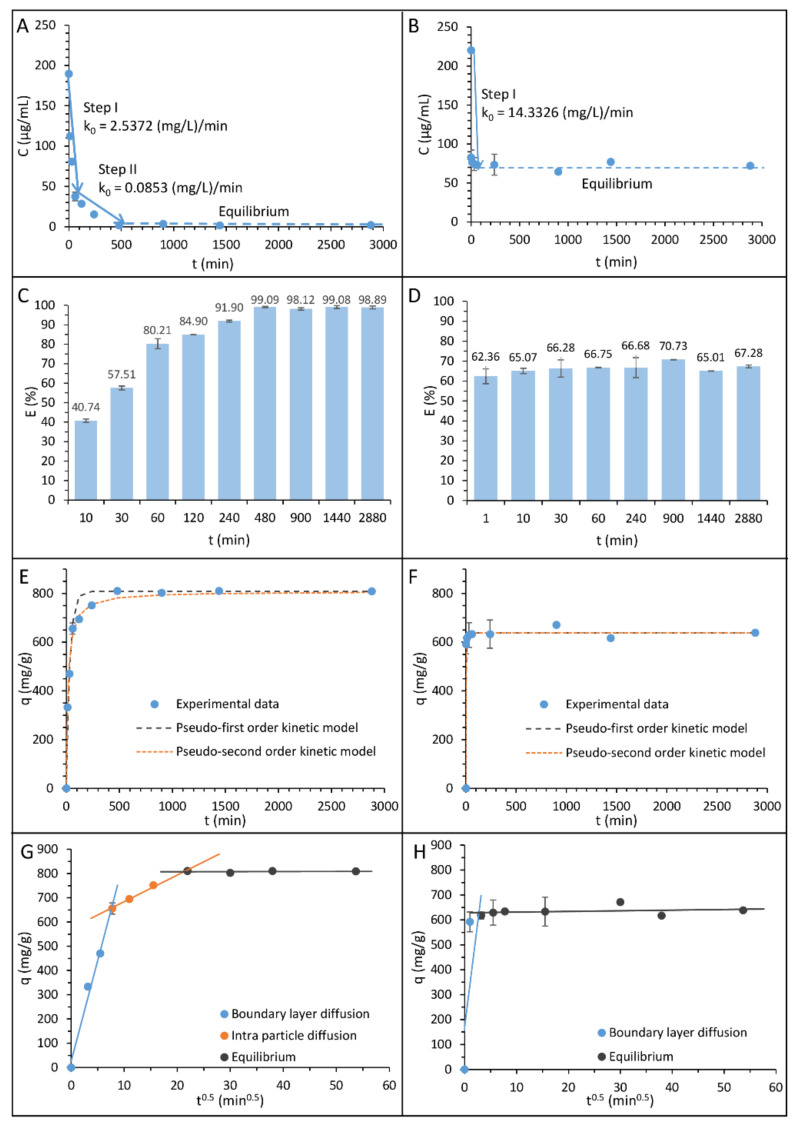
The kinetic steps and rate constant values determined by the zero order kinetic model for ampicillin (**A**) and tetracycline (**B**); the effectiveness of the sorption process for ampicillin (**C**) and tetracycline (**D**); experimental data and the matching of pseudo-first and -second order kinetic models of the ampicillin (**E**) and tetracycline (**F**) sorption by zinc oxide nanoparticles; the Weber–Morris intra-particle diffusion model of ampicillin (**G**) and tetracycline (**H**) sorption by zinc oxide nanoparticles.

**Figure 2 ijms-22-05395-f002:**
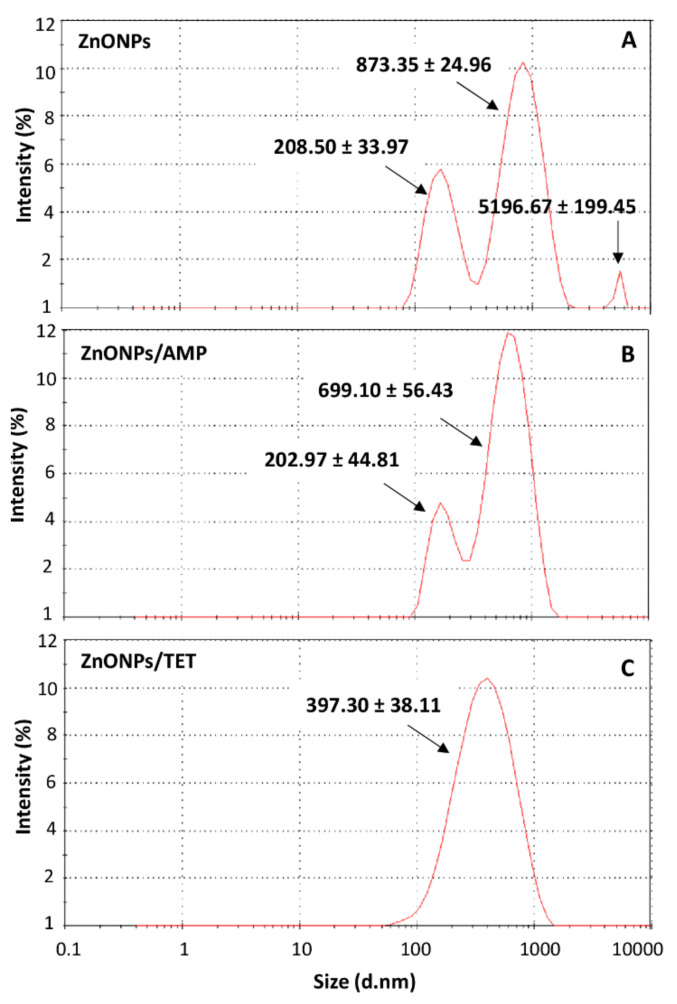
Hydrodynamic size distribution of zinc oxide nanoparticles (**A**), zinc oxide nanoparticles immobilized with ampicillin (**B**) and zinc oxide nanoparticles immobilized with tetracycline (**C**).

**Figure 3 ijms-22-05395-f003:**
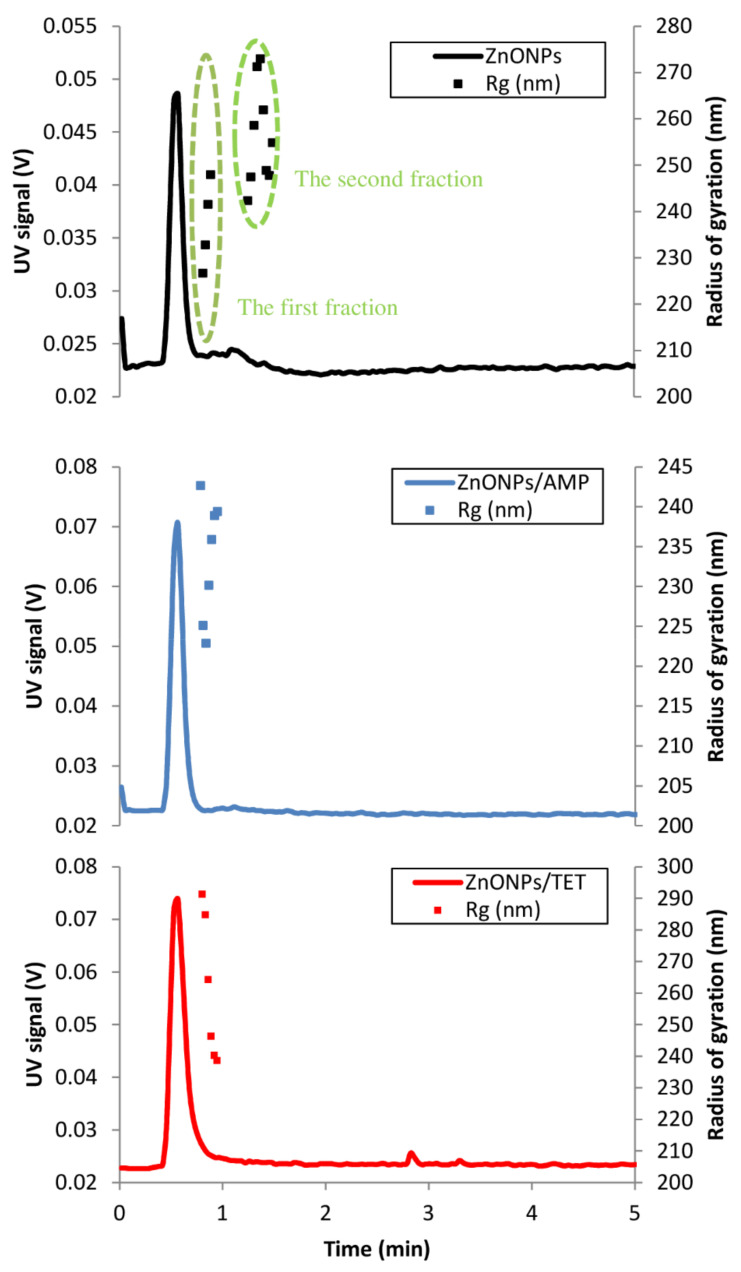
Fractograms of the separation of ZnONPs, ZnONPs/AMP and ZnONPs/TET.

**Figure 4 ijms-22-05395-f004:**
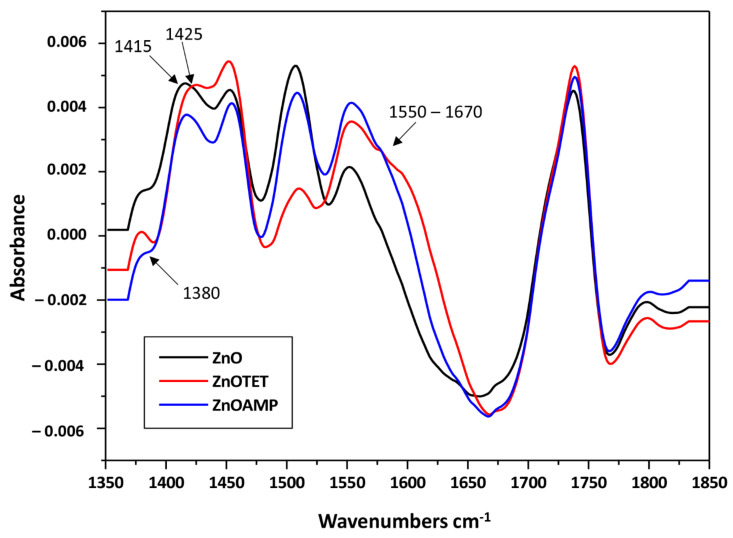
FTIR spectra of non-immobilized and immobilized ZnONPs.

**Figure 5 ijms-22-05395-f005:**
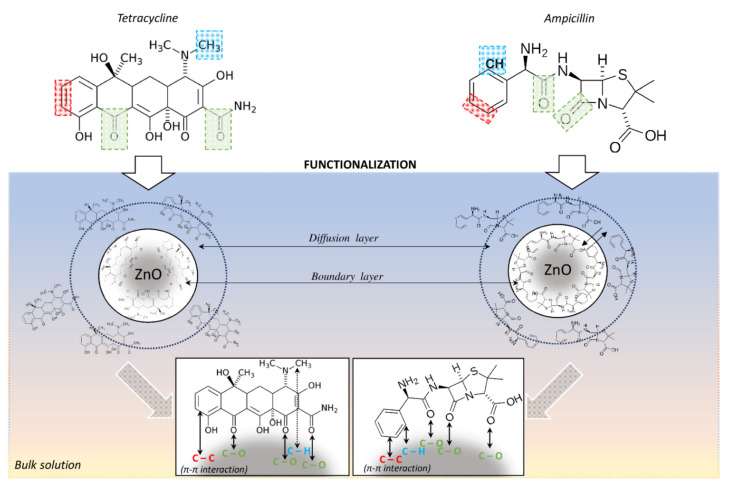
Proposed mechanism of ampicillin and tetracycline binding on the surface of ZnO NPs, where green, blue and red color mark the functional groups of the antibiotics that participate in their interaction with the surface of nanoparticles.

**Figure 6 ijms-22-05395-f006:**
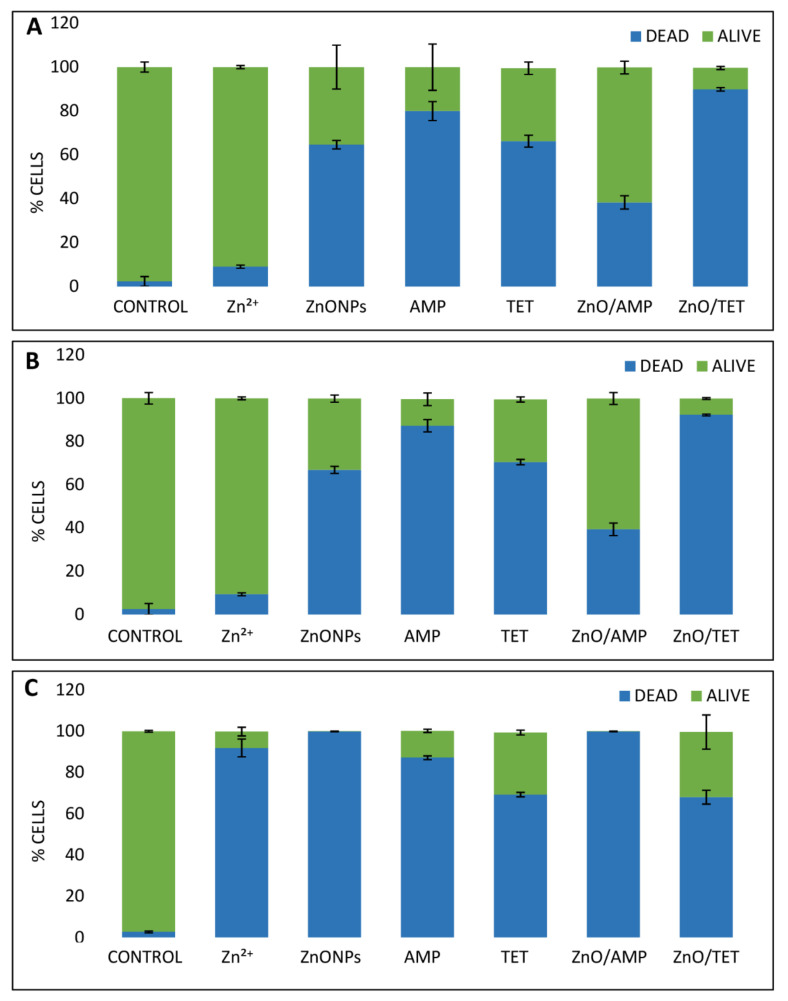
Changes plot of the percentage amount of live and dead *E. coli* (**A**), *S. epidermidis* (**B**) and *K. pneumoniae* (**C**) bacteria cells treated with ampicillin (AMP), tetracycline (TET), zinc ions (Zn^2+^) and zinc oxide nanoparticles (ZnONPs), before and after immobilization with AMP (ZnONPs/AMP) and TET (ZnONPs/TET), as well as non-treated bacteria (control).

**Figure 7 ijms-22-05395-f007:**
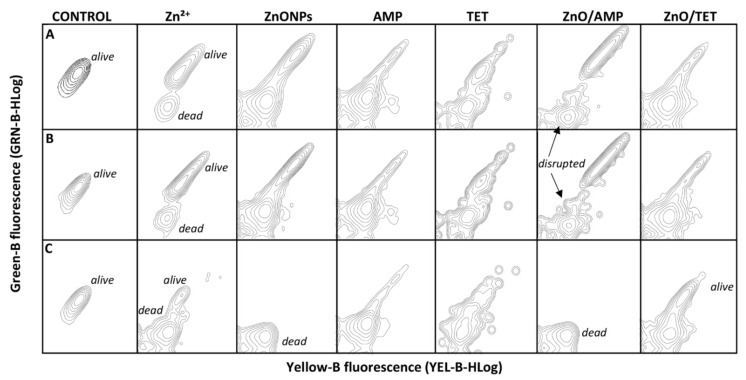
Fluorescence spectra of *E. coli* (**A**), *S. epidermidis* (**B**) and *K. pneumoniae* (**C**) bacteria cells treated with ampicillin (AMP), tetracycline (TET), zinc ions (Zn^2+^) and zinc oxide nanoparticles (ZnONPs), before and after immobilization with AMP (ZnONPs/AMP) and TET (ZnONPs/TET), as well as non-treated bacteria (control).

**Figure 8 ijms-22-05395-f008:**
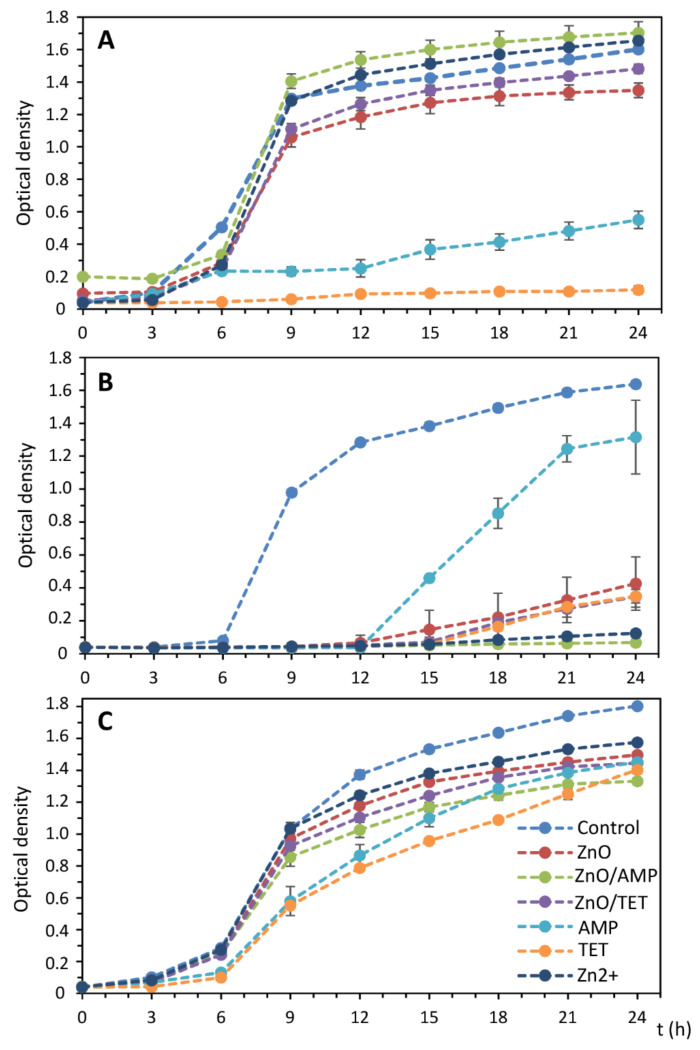
Growth kinetics of non-treated *E. coli* (**A**), *S. epidermidis* (**B**) and *K. pneumoniae* (**C**) bacteria cells, as well as after treatment with ampicillin (AMP), tetracycline (TET), zinc ions (Zn^2+^) and zinc oxide nanoparticles (ZnONPs) before and after immobilization with AMP (ZnONPs/AMP) and TET (ZnONPs/TET), at a concentration of 25% of MIC value.

**Figure 9 ijms-22-05395-f009:**
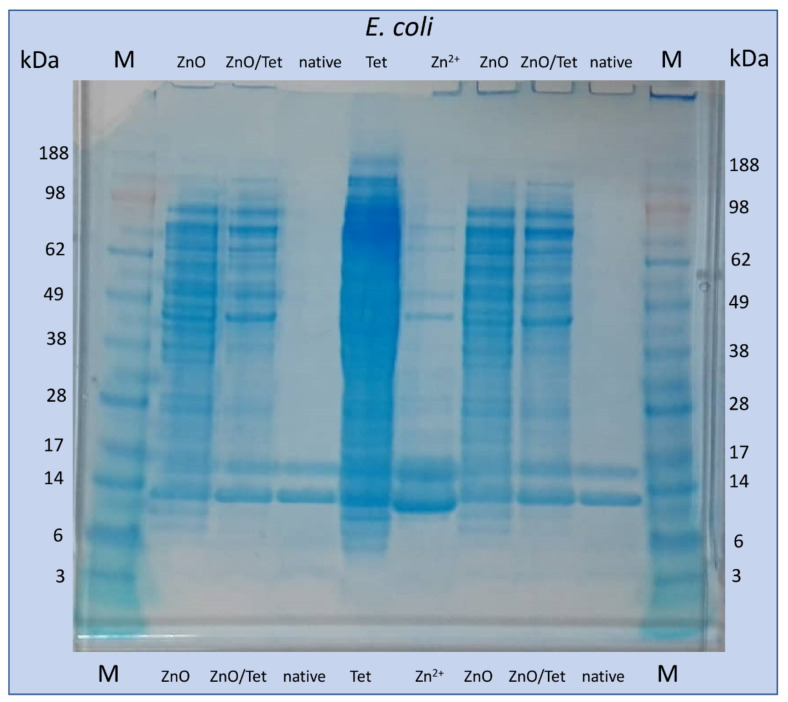
The picture of the gel after electrophoresis of proteins isolated from *E. coli* cells not treated (native) and treated with zinc oxide nanoparticles (ZnO), zinc ions (Zn^2+^), tetracycline (Tet) and zinc oxide nanoparticles, functionalized with tetracycline (ZnO/Tet) and protein marker (M).

**Figure 10 ijms-22-05395-f010:**
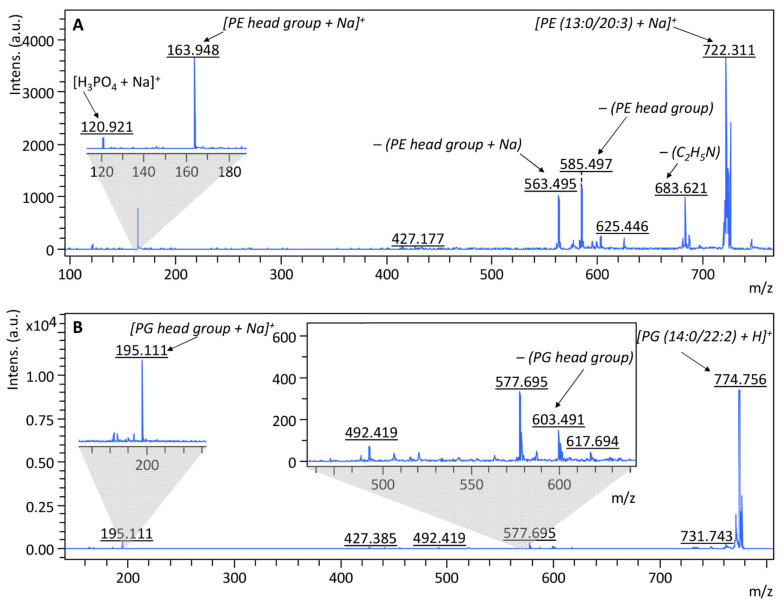
Fragmentation pathway and MS/MS spectra of signal at 722 (**A**) and 775 m/z (**B**).

**Figure 11 ijms-22-05395-f011:**
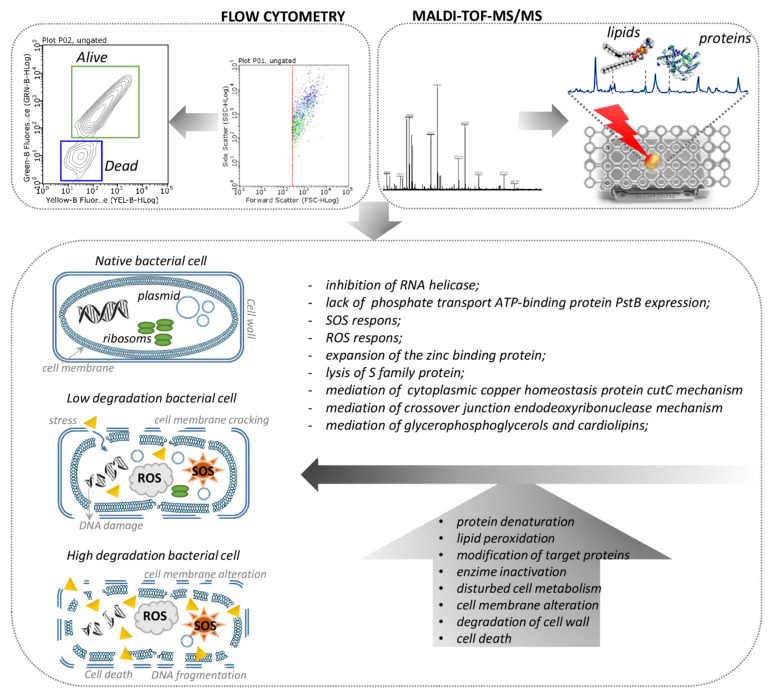
The effects of ZnO NPs and its complexes with ampicillin and tetracycline, along with antibiotics alone on cell metabolism and morphology.

**Figure 12 ijms-22-05395-f012:**
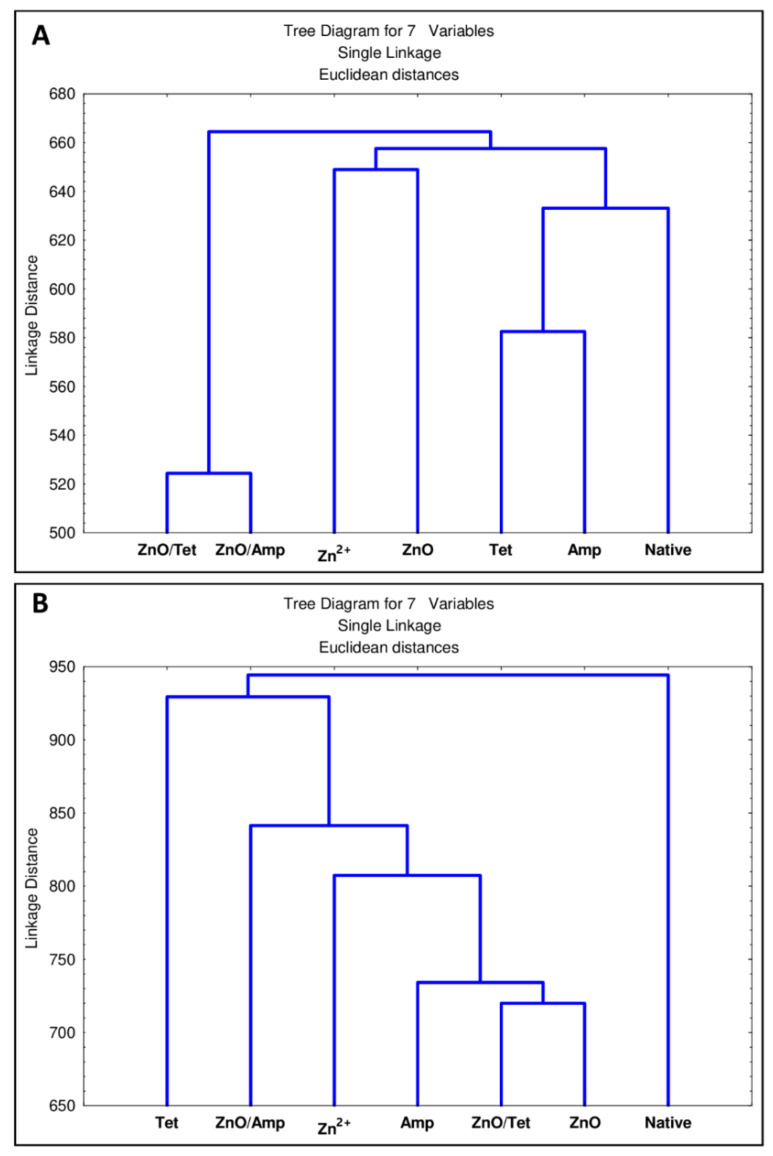
The statistical significance of differences in the protein (**A**) and lipid (**B**) profiles of *E. coli*, *S. epidermidis* and *K. pneumoniae* after their incubation with zinc ions, ampicillin, tetracycline and zinc oxide nanoparticles, before and after immobilization with antibiotics.

**Table 1 ijms-22-05395-t001:** Kinetic model parameters for the ampicillin and tetracycline sorption by zinc oxide nanoparticles.

Zero Order Kinetics Model		Pseudo-First Order Kinetics Model		Pseudo-Second Order Kinetics Model		Intra-Particle Diffusion Model	
**Ampicillin**							
First step							
k_0_ (mg L^−1^ min^−1^)	2.537	q_e_ (mg g^−1^)	808.999	q_e_ (mg g^−1^)	804.298	A (mg g^−1^)	574.577
Second step							
k_0_ (mg L^−1^ min^−1^)	0.085	k_1_ (min^−1^)	0.031	k_2_ (min^−1^)	7.343 × 10^−5^	K_ip_ (mg g^−1^min^−0.5^)	10.959
		A_approx._ %	3.405	A_approx._ %	1.030		
**Tetracycline**							
First step							
k_0_ (mg L^−1^ min^−1^)	14.333	q_e_ (mg g^−1^)	638.784	q_e_ (mg g^−1^)	638.766	A (mg g^−1^)	628.753
		k_1_ (min^−1^)	0.976	k_2_ (min^−1^)	0.019	K_ip_ (mg g^−1^min^−0.5^)	0.263
		A_approx._ %	5.512	A_approx._ %	0.380		

**Table 2 ijms-22-05395-t002:** The values of the distribution coefficient and the change in Gibbs free energy of the ampicillin and tetracycline sorption by zinc oxide nanoparticles.

q_e_ (mg/kg)	C_e_ (mg/L)	K_d_	T (K)	ΔG^0^ (kJ mol^−1^)
**Ampicillin**				
8.1 × 10^5^	2.102	3.8 × 10^5^	295	−31.542
**Tetracycline**				
6.4 × 10^5^	72.077	8.9 × 10^3^	295	−22.293

**Table 3 ijms-22-05395-t003:** The pH, zeta potential, average hydrodynamic size, radius of gyration and polydispersity index of each population value of zinc oxide nanoparticles, non-immobilized and immobilized with ampicillin and tetracycline.

ZnONPs			ZnONPs/AMP		ZnONPs/TET
pH					
7.04			7.01		7.11
Zeta potential (mV)					
−25.23 ± 0.97			−24.57 ± 0.80		−25.17 ± 0.59
Hydrodynamic size (nm)					
**Pk 1**	**Pk 2**	**Pk 3**	**Pk 1**	**Pk 2**	**Pk3**
873.35 ± 24.96	208.50 ± 33.97	5196.67 ± 199.45	699.10 ± 56.43	202.97 ± 44.81	397.30 ± 38.11
Radius of gyration (nm)					
**First fraction**	**Second fraction**		**Fraction**		**Fraction**
237.7	259.0		231.6		263.1
Polydispersity index Pdi					
**First fraction**	**Second fraction**		**Fraction**		**Fraction**
0.99	1.01		0.99		1.01

**Table 4 ijms-22-05395-t004:** The minimum inhibitory concentration (MIC) of ampicillin, tetracycline, zinc ions, zinc oxide nanoparticles immobilized and non-immobilized with antibiotics.

	Minimum Inhibitory Concentration [µg/mL]					
	AMP	TET	Zn^2+^	ZnONPs	ZnONPs/AMP	ZnONPs/TET
*E. coli*	6.25	3.125	300	6.25	6.25	3.125
*S. epidermidis*	50	6.25	150	0.78	1.56	0.78
*K. pneumoniae*	50	0.78	150	1.56	1.56	1.56

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
