# Peer review of "The Study on Molecular Profile Changes of Pathogens via Zinc Nanocomposites Immobilization Approach"

_ijms, 2021, doi:10.3390/ijms22105395_

Round 1

Reviewer 1 Report

In this work, Buszewski and co-workers explored the antibacterial potential of zinc oxide nanoparticles with ampicillin and tetracycline against three bacteria, E. coli, S. epidermidis and K. pneumoniae. Systematic studies were conducted, including the binding between zinc oxide nanoparticles and drugs, minimum inhibitory concentration, flow cytometry and the antibacterial activity mechanism by multiple characterization techniques. However, some introduction information is missing, the experimental data can only partially support the authors’ conclusion. Although a lot of data were obtained, the authors need to put more efforts on data analysis and discussion, to draw more attractive conclusions. English must be improved.

Detailed comments:

  1. Introduction related to antibiotics (tetracycline, ampicillin) is missing. Why do authors choose these antibiotics?
  2. General information related to experiment techniques is missing, for example, the basic concept/background about minimum inhibitory concentration and flow cytometry study, MALDI-TOF.
  3. Figure 1G,H. The unit for x axis should be min0.5.
  4. Line 184, “the Gibbs free energy (ΔG0) and distribution coefficient (Kd) of the process were calculated”. Please add the process (such as the equation ZnO+drug=complex) in the main text. Otherwise, the ΔG is meaningless. In addition, “0” and “d” in ΔG0 and Kd should be subscript.
  5. Table 2. Kd 384825.117 should be “3.8 × 105
  6. Line 192, section 2.2. The authors measured the size distribution of zinc oxide nanopartIcles with or without drugs using DLS, and claimed immobilization caused an increase in homogeneity of nanoparticles. The current data cannot support such statement. In all three cases, the nanoparticles cover the size range 100 – 1000 nm and the distribution difference may be caused by test error. I cannot find any experimental method for DLS and assumed that the scattering angle was set as 90 degree. In order to get more precise data for such large particles, low scattering angle should be used (for example, 30 degree) and CONTIN method should be applied to analyze the obtained correlation function.
  7. Figure 3. For FTIR spectrum of non-immobilized zinc oxide nanoparticles, why does it show so many peaks associated with carbon? The commercially available zinc oxide should contain no carbons.
  8. What is the nature of binding between zinc oxide and the drugs? Hydrophobic interactions? Hydrogen bonding?
  9. Figure 3. The absorbance should not be negative.
  10. Figure 4. Chemical structures in the lower positions. Please clarify the use of bonds (C-H, C=O, C-C).
  11. Line 226. “However, the mechanism of ZnONPs/AMP action was different in comparison to non-immobilized ZnONPs”. How was this conclusion obtained? What is the difference? More conclusion should be addressed based on current data.
  12. The authors used MS to analyze possible protein fragments with low molecular weights. How about those proteins with high molecular weights, as shown in Figure 8?
  13. English should be improved. For example, the use of “in turn”.

Author Response

Response to Reviewer 1 Comments

We would like to thank the Editor and Reviewers for careful reading, and constructive suggestions for our manuscript that will help us to improve our work. According to this, we have comprehensively revised our manuscript. Hoping that we have addressed all the questions mentioned by the Reviewers, below we included the point-to-point response to each comments of Reviewer.

In manuscript file all of the changes have been provided in red color.

REVIEWER COMMENTS:

In this work, Buszewski and co-workers explored the antibacterial potential of zinc oxide nanoparticles with ampicillin and tetracycline against three bacteria, E. coli, S. epidermidis and K. pneumoniae. Systematic studies were conducted, including the binding between zinc oxide nanoparticles and drugs, minimum inhibitory concentration, flow cytometry and the antibacterial activity mechanism by multiple characterization techniques. However, some introduction information is missing, the experimental data can only partially support the authors’ conclusion. Although a lot of data were obtained, the authors need to put more efforts on data analysis and discussion, to draw more attractive conclusions. English must be improved.

Point 1: Introduction related to antibiotics (tetracycline, ampicillin) is missing. Why do authors choose these antibiotics?

Response 1: According to the literature data, it is assumed that both, ampicillin and tetracycline are still the first-line antibiotics in in the treatment of bacterial infections: throat, skin, especially difficult to heal wounds. Although the known ampicillin/ tetracycline resistance of pathogenic strains, the listed drugs are unreasonably overused and it is unmet need to investigate them in the context of searching for a new type of antiseptics. According to the previous work published by our group [1–3], it was proven that the combination of Amp/Tet with nanoparticles increase their antibacterial activity and can be considered as a new type of antiseptics. Therefore, the present research was focused on the same antibiotics but different nanoparticles. Taking into consideration the Reviewer suggestion, the Introduction section is supplemented by information related to the choose of antibiotics.

Point 2: General information related to experiment techniques is missing, for example, the basic concept/background about minimum inhibitory concentration and flow cytometry study, MALDI-TOF.

Response 2: We would like to thank the Reviewer for this comment. Commonly, the assays such as minimal inhibitory concentration (MIC) or colony forming unit (CFU) are used for the biological activity of nanoparticles/antibiotics investigation. Recently, the new and more precise techniques arouse the interest of scientists. Therefore, in present study, has been  applied interdisciplinary approach including both, traditional method (MIC) as well as complementary experimental techniques (flow cytometry, spectroscopy and mass spectrometry). Moreover, the proposed approach allow to connect physiological changes in bacteria cells after ZnO NPs/antibiotic treatment not only at the cellular but also molecular level. Therefore, this aspect it extend the knowledge about the antibacterial activity mechanism of the investigated ZnO complexes.

According to the Reviewer’s remark the Introduction section is supplemented by the basic concept of using this techniques. 

Point 3: Figure 1G,H. The unit for x axis should be min0.5.

Response 3: We would like to thank Reviewer for such careful revision of the manuscript. This mistake has been corrected and Figure 1 was appropriate modified.

Point 4: Line 184, “the Gibbs free energy (ΔG0) and distribution coefficient (Kd) of the process were calculated”. Please add the process (such as the equation ZnO+drug=complex) in the main text. Otherwise, the ΔG is meaningless. In addition, “0” and “d” in ΔG0 and Kd should be subscript.

Response 4: We would like to apologize for this mistake. The indexes have been corrected and a corresponding explanation has been added in the manuscript.

Point 5: Table 2. Kd 384825.117 should be “3.8 × 105

Response 5: Taking in consideration the Reviewer suggestion the data was modified and presented in a clearer form.

Point 6: Line 192, section 2.2. The authors measured the size distribution of zinc oxide nanopartIcles with or without drugs using DLS, and claimed immobilization caused an increase in homogeneity of nanoparticles. The current data cannot support such statement. In all three cases, the nanoparticles cover the size range 100 – 1000 nm and the distribution difference may be caused by test error. I cannot find any experimental method for DLS and assumed that the scattering angle was set as 90 degree. In order to get more precise data for such large particles, low scattering angle should be used (for example, 30 degree) and CONTIN method should be applied to analyze the obtained correlation function.

Response 6: We would like to thank the Reviewer for this valuable remark. The Reviewer is right, based on the obtained results we cannot claim that immobilization caused an increase in homogeneity of nanoparticles. Zetasizer used in present study (Zetasizer ZS, Malvern Instruments, Malvern, Great Britain) determine the size of particles measuring firstly, Brownian motion of particles using Dynamic Light Scattering (DLS) and then convert it to particle size distribution using the Stokes-Einstein equation. According, to the ZETAsizer ZS guidelines the technique is allowed to measure the size of particles from 0.6 nm to 6 µm at  173° measurement angle.   Therefore, the obtained results indicate on the polydisperse nature of the hydrodynamic radius distribution of nanoparticles before and after  functionalization, explaining also the aggregation ability of the particles. Functionalization reduces this polydispersity, and in the case of tetracycline-functionalized nanoparticles, the distribution of the hydrodynamic radius obtained the Gaussian character. An appropriate explanation has been added in the main text.

Point 7: Figure 3. For FTIR spectrum of non-immobilized zinc oxide nanoparticles, why does it show so many peaks associated with carbon? The commercially available zinc oxide should contain no carbons.

Response 7: We would like to thank the Reviewer for this valuable remark. As the Reviewer rightly noted, commercially available zinc oxide nanoparticles should not contain carbon, especially since the manufacturer did not provide information about the presence of a stabilizer. However, the presence of specific signals in the FTIR spectra indicates the presence of carbon in the sample. This element may be a residue from the synthesis process or come from a stabilizer. To confirm the presence of carbon in the purchased nanoparticles, we decided to conduct an additional experiment. TOC analysis (Fig. 1) showed that the sample contains a carbon (TC ~ 60 mg/L). To avoid confusion, we have added this information to the manuscript in the section "Physico-chemical characteristics of immobilized zinc oxide nanoparticles".

Fig. 1. The results of TOC analysis of zinc oxide nanoparticles.

Point 8: What is the nature of binding between zinc oxide and the drugs? Hydrophobic interactions? Hydrogen bonding?

Response 8: We postulate that the interactions between the zinc oxide nanoparticle and the antibiotic are of a mixed nature, and that hydrophobic interactions have a decisive share in the binding due to the hydrophobic nature of zinc oxide. However, ZnO solvation and the share of water in indirect interactions should also be taken into account which was confirmed by our previous study on Zn interactions witch organic ligands [4]. To accurately determine the type of ZnONPs-drug interactions, it will be necessary to conduct additionally research using molecular modeling, which we plan in the future. The manuscript has been supplemented with an appropriate explanation.

Point 9: Figure 3. The absorbance should not be negative.

Response 9: We thanks the Reviewer for the valuable remark.  According to the technical guide regarding the analysis method for direct detect the software subtracts buffer signal from only a part of the spectrum, resulting in partially processed spectrum covering the region where subtraction has been applied. In the next step, the partially processed spectrum is integrated. The software anchors a baseline, that runs parallel to the x axis, at a basepoint outside the exact analysis region  and determines strength of the amide signal at the predetermined wavenumber. The authors hope that the Reviewer will find this answer justified.

Point10: Figure 4. Chemical structures in the lower positions. Please clarify the use of bonds (C-H, C=O, C-C).

Response 10: The authors used the above-mentioned bonds to show which functional groups of the antibiotic participate in their interaction with the surface of nanoparticles. We agree with the Reviewer that this may not be clear to the reader, so we decided to revised and improve the caption of Figure 4.

Point 11: Line 226. “However, the mechanism of ZnONPs/AMP action was different in comparison to non-immobilized ZnONPs”. How was this conclusion obtained? What is the difference? More conclusion should be addressed based on current data.

Response 11: We agree with the Reviewer that this conclusion requires a more detailed explanation. The given conclusion is related with differences in the observed fluorescence spectra. In the case of unmodified nanoparticles, signals from live and dead cells were observed, while the functionalization of ZnONPs with ampicillin resulted in the appearance of broken cell fragments. The manuscript has been supplemented with this information.

Point 12: The authors used MS to analyze possible protein fragments with low molecular weights. How about those proteins with high molecular weights, as shown in Figure 8?

Response 12: We would like to thank the Reviewer for paying attention to this issue. Research using gel electrophoresis in one dimension (1D) was aimed to obtain the first information on whether the incubation of bacterial cells with stressors significantly influences on changes in the overall protein profile. We used this method as a complementary method to the MALDI technique. In turn, the MALDI analysis carried out in a linear mode was aimed to identify the molecular protein profile of cells. In the case of MS analysis, the post source decay of high protein complex, resulting in no signals from high mass proteins. In turn, in the case of 1D electrophoresis analysis, the native protein profile was examined. In the future, we plan to use 2D electrophoresis and further protein analysis using the bottom-up method using the MALDI technique. An appropriate explanation has been added in the main text.

Point 13: English should be improved. For example, the use of “in turn”.

Response 13: According to Reviewer suggestion the manuscript was revised and edited by native English speaker.

References

  1. Buszewski, B.; RafiÅ„ska, K.; Pomastowski, P.; Walczak, J.; Rogowska, A. Novel aspects of silver nanoparticles functionalization. Colloids Surfaces A Physicochem. Eng. Asp. 2016, 506, 170–178, doi:10.1016/J.COLSURFA.2016.05.058.
  2. Rogowska, A.; RafiÅ„ska, K.; Pomastowski, P.; Walczak, J.; Railean-Plugaru, V.; Buszewska-Forajta, M.; Buszewski, B. Silver nanoparticles functionalized with ampicillin. Electrophoresis 2017, 38, 2757–2764, doi:10.1002/elps.201700093.
  3. Buszewski, B.; Rogowska, A.; Railean-Plugaru, V.; Złoch, M.; Walczak-Skierska, J.; Pomastowski, P. The Influence of Different Forms of Silver on Selected Pathogenic Bacteria. Materials (Basel). 2020, 13, 2403, doi:10.3390/ma13102403.
  4. Buszewski, B.; Žuvela, P.; Król-Górniak, A.; Railean-Plugaru, V.; Rogowska, A.; Wong, M.W.; Yi, M.; Rodzik, A.; Sprynskyy, M.; Pomastowski, P. Interactions of zinc aqua complexes with ovalbumin at the forefront of the Zn2+/ZnO-OVO hybrid complex formation mechanism. Appl. Surf. Sci. 2021, 542, 148641, doi:10.1016/j.apsusc.2020.148641.
  5. Elshikh, M.; Ahmed, S.; Funston, S.; Dunlop, P.; McGaw, M.; Marchant, R.; Banat, I.M. Resazurin-based 96-well plate microdilution method for the determination of minimum inhibitory concentration of biosurfactants. Biotechnol. Lett. 2016, 38, 1015–1019, doi:10.1007/s10529-016-2079-2.

Reviewer 2 Report

Dear Authors.

The article entitled The study on molecular profile changes of pathogens via zinc nanocomposites immobilization approach is a very comprehensive study on strategy of using ZnO NPs immobilization with ampicillin and tetracycline. The answer to the research hypothesis was achieved by application of several laboratory methods including determination of immobilization kinetics, physico-chemical characterization, antimicrobial activity determination (classical broth-microdilution method, flow cytometry, growth kinetics), MALDI-TOF MS analyses and gel electrophoreses.

For the majority of the methods used, interpretation of the results as well as discussion I have no objections.

However, the major concern is the language style. I strictly recommend the article to be revised by Native speaking specialist. Few examples:

  • The bacterial species names should be written in Italics
  • Full Species and generic names should be given when used initially, for instance Staphyclococcus aureus......-> S. aureus.
  • K. pneumoniae not K.pneumonia
  • Mueller-Hinton Broth/Medium not Miller Hinton Medium
  • Many linguistic errors (also in abstract) and colloquialisms.

Despite this fact I have several considerations that also need to be revised:

  1. There is no justification for the use of ampicillin and tetracycline. Please explain.
  2. Why S. epidermidisE. coli and K. pneumoniae where chosen for antimicrobial assays? Please explain. Especially when we are dealing with S. epidermidis, K. penumoanie and ampicillin. Since the majority of coagulase-negative staphylococci (S. epidermidis) are penicillinase producers so they are resistant to benzylpenicillin, phenoxymethylpenicillin, ampicillin, amoxicillin, piperacillin and ticarcillin. Moreover, K. pneumoniae usually shows intrinsic resistance to ampicillin and other β-lactams. 
  3. There is lack of information about the source of S. epidermidis strain. As I understand, all strains were obtained from the collection of Centre for Modern Interdisciplinary Technologies, Nicolaus Copernicus? But. Does S. epidermidis was reference or clinical strain? If reference, the proper number, name of collection should be given.
  4. What method was applied in the case of MIC determination? Was it broth microdilution or macrodilution method (Both recommended by CLSI)? This should be specified. If microtitter plates were used, then the type of material (PP or PS) should be given
  5. What was the concentration of resazurin and time of incubation (temperature) with the cell viability agent? Please revise
  6. For CLSI method, MIC values should be determined visually (without any dyes). That is why, a proper reference should given additionally.
  7. Did Authors checked the impact of Zn and ZOPs etc. on resazurin alone? Please explain
  8. Resazurin based readings can be achieved spectrophotometrically. Why visual determination was done? Please explain
  9. Line 549. For flow cytometry analysis the samples were prepared in the same way as it was described in section 2.5. However, section 2.5 is in Results Section (Electrophoretic analysis...)
  10. The same for Line 573. There is no section 2.7

The last 2 points indicate that the article should be once again reviewed also by the Authors to avoid such trival technical errors. The substantive and linguistic (English) analysis is also needed. Then the article can be considered for publication.

Author Response

Response to Reviewer 2 Comments

We would like to thank the Editor and Reviewers for careful reading, and constructive suggestions for our manuscript that will help us to improve our work. According to this, we have comprehensively revised our manuscript. Hoping that we have addressed all the questions mentioned by the Reviewers, below we included the point-to-point response to each comments of Reviewer.

In manuscript file all of the changes have been provided in red color.

REVIEWER COMMENTS:

Dear Authors.

The article entitled The study on molecular profile changes of pathogens via zinc nanocomposites immobilization approach is a very comprehensive study on strategy of using ZnO NPs immobilization with ampicillin and tetracycline. The answer to the research hypothesis was achieved by application of several laboratory methods including determination of immobilization kinetics, physico-chemical characterization, antimicrobial activity determination (classical broth-microdilution method, flow cytometry, growth kinetics), MALDI-TOF MS analyses and gel electrophoreses.

For the majority of the methods used, interpretation of the results as well as discussion I have no objections.

Point 1: However, the major concern is the language style. I strictly recommend the article to be revised by Native speaking specialist. Few examples:

  • The bacterial species names should be written in Italics
  • Full Species and generic names should be given when used initially, for instance Staphyclococcus aureus......-> S. aureus.
  • K. pneumoniae not K.pneumonia
  • Mueller-Hinton Broth/Medium not Miller Hinton Medium
  • Many linguistic errors (also in abstract) and colloquialisms.

Response 1: We would like to apologize for all of these mistakes. Considering all of the above mentioned by Reviewer comments all off mistakes were corrected. Moreover, all manuscript was once again revised to avoid any typos or grammatical errors. In addition, the manuscript was edited by native English specialist. We hope that the made corrections will be satisfactory.

Despite this fact I have several considerations that also need to be revised:

Point 2: There is no justification for the use of ampicillin and tetracycline. Please explain.

Response 2: According to the literature data, it is assumed that both, ampicillin and tetracycline are still the first-line antibiotics in the treatment of bacterial infections: throat, skin, especially difficult to heal wounds. Although the known ampicillin/ tetracycline resistance of pathogenic strains, the listed drugs are unreasonably overused and it is unmet need to investigate them in the context of searching for a new type of antiseptics. Moreover, based on our previous work [1–3], it was proven that the combination of Amp/Tet with nanoparticles increase their antibacterial activity.

Taking into consideration the Reviewer suggestion, the Introduction section is supplemented by information related to the choose of antibiotics.

Point 3: Why S. epidermidisE. coli and K. pneumoniae where chosen for antimicrobial assays? Please explain. Especially when we are dealing with S. epidermidis, K. penumoanie and ampicillin. Since the majority of coagulase-negative staphylococci (S. epidermidis) are penicillinase producers so they are resistant to benzylpenicillin, phenoxymethylpenicillin, ampicillin, amoxicillin, piperacillin and ticarcillin. Moreover, K. pneumoniae usually shows intrinsic resistance to ampicillin and other β-lactams. 

Response 3: We would like to thank the Reviewer for a such valuable comment. We agree that the bacterial strains used in our study are known as a resistant for ampicillin. Although, both, ampicillin and tetracycline are still the first-line drugs and they are overdose in treatment of  many infections, even those for which this group of antibiotics does not work (data from European Centre for Disease Prevention and Control). Taking into consideration this fact, therefore, the known ampicillin/tetracycline-resistance of pathogenic strains, it is necessary to study these strains and antibiotics in the context of searching for a new type of antiseptics.

Point 4: There is lack of information about the source of S. epidermidis strain. As I understand, all strains were obtained from the collection of Centre for Modern Interdisciplinary Technologies, Nicolaus Copernicus? But. Does S. epidermidis was reference or clinical strain? If reference, the proper number, name of collection should be given.

Response 4: Escherichia coli ATCC25922 and Klebsiella pneumoniae ATCC BAA-1144 was purchased from Pol-Aura (Dywity, Poland). In turn, Staphylococcus epidermidis was previously isolated from honey by our research group (accession number MH045861). According to Reviewer suggestion the “Materials and methods” section has been supplemented with missing information.

Point 5: What method was applied in the case of MIC determination? Was it broth microdilution or macrodilution method (Both recommended by CLSI)? This should be specified. If microtitter plates were used, then the type of material (PP or PS) should be given

Response 5: In the present study, the MIC value was determined based on the  broth microdilution method. For the analysis the 96-Well Cell Culture Plates (PP)  have been used. . According to Reviewer remark the “Materials and methods” section has been supplemented with missing information.

Point 6: What was the concentration of resazurin and time of incubation (temperature) with the cell viability agent? Please revise

Response 6: We would like to thank the Reviewer for paying attention to this issue. The final concentration of resazurin in each well was 45,8 μL/mL and the time of incubation was 24 h in 37oC. According to Reviewer suggestion the “Materials and methods” section has been revised and supplemented with missing information in order to avoid the misunderstanding.

Point 7: For CLSI method, MIC values should be determined visually (without any dyes). That is why, a proper reference should given additionally.

Response 7: According to Reviewer suggestion we added the appropriate citation in the “Materials and methods” section.

Point 8: Did Authors checked the impact of Zn and ZOPs etc. on resazurin alone? Please explain

Response 8: We would like to thank the Reviewer for this question. During the tests, a negative control was carried out in which a high concentration (100 µg/mL) of unmodified and modified zinc oxide nanoparticles in Mueller-Hinton Broth medium was used. Samples were incubated for 24 h in the dark at 37 °C. Figure 1 below shows the obtained results. No influence of nanoparticles on the color of resazurin under the experimental conditions was observed. Relevant information has been added in the section of materials and methods in manuscript.

Fig. 1. The negative control for MIC experiments.

Point 9: Resazurin based readings can be achieved spectrophotometrically. Why visual determination was done? Please explain

Response 9: According to the literature data, the resozurine assay is performed in both ways: using spectrophotometric approach and also using visual method based on resazurin dye coloration [5].  The lowest concentration at which no change in color can be observed is considered as the MIC value. Moreover, as the color change was very distinct and visible to the naked eye, no spectrophotometric reading was needed. According to the Reviewer remark the material and method section was revised and improved. 

Point 10: Line 549. For flow cytometry analysis the samples were prepared in the same way as it was described in section 2.5. However, section 2.5 is in Results Section (Electrophoretic analysis...)

The same for Line 573. There is no section 2.7

Response 10: We would like to thank Reviewer for such careful revision of the manuscript. This mistake has been corrected.

The last 2 points indicate that the article should be once again reviewed also by the Authors to avoid such trival technical errors. The substantive and linguistic (English) analysis is also needed. Then the article can be considered for publication.

References

  1. Buszewski, B.; RafiÅ„ska, K.; Pomastowski, P.; Walczak, J.; Rogowska, A. Novel aspects of silver nanoparticles functionalization. Colloids Surfaces A Physicochem. Eng. Asp. 2016, 506, 170–178, doi:10.1016/J.COLSURFA.2016.05.058.
  2. Rogowska, A.; RafiÅ„ska, K.; Pomastowski, P.; Walczak, J.; Railean-Plugaru, V.; Buszewska-Forajta, M.; Buszewski, B. Silver nanoparticles functionalized with ampicillin. Electrophoresis 2017, 38, 2757–2764, doi:10.1002/elps.201700093.
  3. Buszewski, B.; Rogowska, A.; Railean-Plugaru, V.; Złoch, M.; Walczak-Skierska, J.; Pomastowski, P. The Influence of Different Forms of Silver on Selected Pathogenic Bacteria. Materials (Basel). 2020, 13, 2403, doi:10.3390/ma13102403.
  4. Buszewski, B.; Žuvela, P.; Król-Górniak, A.; Railean-Plugaru, V.; Rogowska, A.; Wong, M.W.; Yi, M.; Rodzik, A.; Sprynskyy, M.; Pomastowski, P. Interactions of zinc aqua complexes with ovalbumin at the forefront of the Zn2+/ZnO-OVO hybrid complex formation mechanism. Appl. Surf. Sci. 2021, 542, 148641, doi:10.1016/j.apsusc.2020.148641.
  5. Elshikh, M.; Ahmed, S.; Funston, S.; Dunlop, P.; McGaw, M.; Marchant, R.; Banat, I.M. Resazurin-based 96-well plate microdilution method for the determination of minimum inhibitory concentration of biosurfactants. Biotechnol. Lett. 2016, 38, 1015–1019, doi:10.1007/s10529-016-2079-2.

Round 2

Reviewer 1 Report

I believe the authors have responded to most comments accordingly. I would recommend publishing this manuscript if the authors can address the two points below.

1. Point 6 and response 6, Figure 2 in main text. I agree with the authors that “the obtained results indicate on the polydisperse nature of the hydrodynamic radius distribution of nanoparticles before and after functionalization”, however, single measurement results at high scattering angle (173 degree in this case) are not strong evidence. One limitation from DLS is low resolution, especially when sizes of two species present in the solution are close (less than a factor of 3), DLS cannot precisely characterize a polydisperse sample, where a broad peak instead of two separate peaks will be observed. In this case, repeat measurements or measurements at multiple scattering angles, and CONTIN analysis are needed in order to give such conclusion. One suggestion is that the authors can compare the original data (correlation function) of these three measurements to check whether the three measurement results are distinguished from others.

Considering this conclusion is not influencing much the main conclusion, I would suggest deleting such statement.

2. Point 9. Please clarify it in the manuscript, otherwise it may cause confusion for readers.

Author Response

We would like to thank the Reviewer for careful reading, and constructive suggestions for our manuscript that will help us to improve our work. According to this, we have comprehensively revised our manuscript. Hoping that we have addressed all the questions mentioned by the Reviewer, in attatched file we included the point-to-point response to each comments of Reviewer.

In manuscript file all of the changes have been provided in red color.

Reviewer 2 Report

Dear Authors.

The majority of considerations have been addressed. 

But, after the 1st Round of the Review I have some considerations concerning MIC determination.

I agree that resazurin can be used in MIC determination, especially in the case of some compounds. However, Authors indicate that "the time of incubation was 24 h in 37oC" (Response 6).

I cannot accept this explanation.

This time is much to long, especially for Gram-negative bacteria. Even if their growth was inhibited, the remaining bacteria can lead to noticeable reduction of resazurin during 24-hour incubation with stirring.

Authors referred to the method of Elshikh et al. Biotechnol Lett. 2016; 38: 1015–1019. But, in this work:

After incubation for 24 h at 37 °C, resazurin (0.015 %) was added to all wells (30 µl per well), and further incubated for 2–4 h for the observation of colour change. On completion of the incubation, columns with no colour change (blue resazurin colour remained unchanged) were scored as above the MIC value.

The time of 2-4 h is in line with my own experiences. When Authors base on particular works, then the protocol should follow all steps of the procedure exactly.

This should be explained, because the results of microbiological studies affected other assays in this paper. This means that if the microbiological assays need to be repeated then the majority of the work should be repeated as well.

In my opinion this fact may explain a really high concentration of AMP in the case of S. epidermidis since concentrations > 16μg/mL are very rare for clinical strains (Please see https://mic.eucast.org/search/?search%5Bmethod%5D=mic&search%5Bantibiotic%5D=-1&search%5Bspecies%5D=461&search%5Bdisk_content%5D=-1&search%5Blimit%5D=50)

For me, it is only objection but needs proper explanation.

Author Response

We would like to thank the Reviewer for careful reading, and constructive suggestions for our manuscript that will help us to improve our work. According to this, we have comprehensively revised our manuscript. Hoping that we have addressed all the questions mentioned by the Reviewer, in attatched file we included the point-to-point response to each comments of Reviewer.

In manuscript file all of the changes have been provided in red color.

Point 1: I agree that resazurin can be used in MIC determination, especially in the case of some compounds. However, Authors indicate that "the time of incubation was 24 h in 37oC" (Response 6).

I cannot accept this explanation.

This time is much to long, especially for Gram-negative bacteria. Even if their growth was inhibited, the remaining bacteria can lead to noticeable reduction of resazurin during 24-hour incubation with stirring.

Authors referred to the method of Elshikh et al. Biotechnol Lett. 2016; 38: 1015–1019. But, in this work:

After incubation for 24 h at 37 °C, resazurin (0.015 %) was added to all wells (30 µl per well), and further incubated for 2–4 h for the observation of colour change. On completion of the incubation, columns with no colour change (blue resazurin colour remained unchanged) were scored as above the MIC value.

The time of 2-4 h is in line with my own experiences. When Authors base on particular works, then the protocol should follow all steps of the procedure exactly.

This should be explained, because the results of microbiological studies affected other assays in this paper. This means that if the microbiological assays need to be repeated then the majority of the work should be repeated as well.

In my opinion this fact may explain a really high concentration of AMP in the case of S. epidermidis since concentrations > 16μg/mL are very rare for clinical strains (Please see https://mic.eucast.org/search/?search%5Bmethod%5D=mic&search%5Bantibiotic%5D=1&search%5Bspecies%5D=461&search%5Bdisk_content%5D=1&search%5Blimit%5D=50)

For me, it is only objection but needs proper explanation.

Response 1: We would like to emphasize that we have deliberately applied such a long incubation time. As the Reviewer rightly noted such long time of incubation enabling for the noticed reduction of resazurin even if only small amounts of remaining bacterial cells was present in the sample. This allowed for a better modeling of the clinical effect of the tested antibacterial agents. In addition, according to research by Sarker et al. (2007) resazurin can be added at the beginning of the 24-hour incubation [1]. Regarding the work of Elshikha et al. (2016) [2], we referred to it only in the context of visually reading of the results based on the color change of the indicator, not to the entire procedure. We apologize for this confusion. The description of methodological part has been revised.

On the other hand, the high concentration of AMP in the case of S. epidermidis is, in our opinion, related to the origin of the tested strain. Since this bacterium was isolated from environmental samples, it may behave in a different way than bacteria isolated from clinical samples. S. epidermidis used in our study was isolated from honey, so attention should be paid to the influence of the extreme environment (high content of sugars and natural antibacterial compounds, presence the hydrogen peroxide, low pH, high osmotic pressure) on its drug susceptibility [3]. It has been proven that bacterial growth in extreme environments can lead to increased bacterial resistance to commercial drugs [4]. Therefore, the high concentration of ampicillin necessary to inhibit the growth of this bacterium may be related to the source of its origin and we cannot compare this result with the results obtained for the same strain but of a different origin.

To avoid misunderstandings, a suitable explanation has been added in main text.

  1. Sarker, S.D.; Nahar, L.; Kumarasamy, Y. Microtitre plate-based antibacterial assay incorporating resazurin as an indicator of cell growth, and its application in the in vitro antibacterial screening of phytochemicals. Methods 2007, 42, 321–324, doi:10.1016/j.ymeth.2007.01.006.
  2. Elshikh, M.; Ahmed, S.; Funston, S.; Dunlop, P.; McGaw, M.; Marchant, R.; Banat, I.M. Resazurin-based 96-well plate microdilution method for the determination of minimum inhibitory concentration of biosurfactants. Biotechnol. Lett. 2016, 38, 1015–1019, doi:10.1007/s10529-016-2079-2.
  3. Silva, M.S.; Rabadzhiev, Y.; Eller, M.R.; Iliev, I.; Ivanova, I.; Santana, W.C. Microorganisms in Honey. In Honey Analysis; InTech, 2017.
  4. McMahon, M.A.S.; Xu, J.; Moore, J.E.; Blair, I.S.; McDowell, D.A. Environmental stress and antibiotic resistance in food-related pathogens. Appl. Environ. Microbiol. 2007, 73, 211–217, doi:10.1128/AEM.00578-06.

Round 3

Reviewer 2 Report

Dear Authors,

All considerations have been adressed.

The manuscript can be accepted for publication

Best regards